# Transformer-Modulated Diffusion Models for Probabilistic Multivariate Time Series Forecasting

**Yuxin Li**[1], **Wenchao Chen**[*, 1], **Xinyue Hu**[1], **Bo Chen**[*, 1] , **Baolin Sun**[2], **Mingyuan Zhou**[3]

[1] National Key Laboratory of Radar Signal Processing, Xidian University, Xi'an, 710071, China.
[2] Xidian University, Xi'an, 710071, China.
[3] McCombs School of Business, The University of Texas at Austin, Austin, TX 78712
`yuxinli2020@gmail.com  wcchen_xidian@163.com  xinyuehu122@gmail.com`
`bchen@mail.xidian.edu.cn  sunbaolin0903@gmail.com`
`mingyuan.zhou@mccombs.utexas.edu`

## Abstract

Transformers have gained widespread usage in multivariate time series (MTS) forecasting, delivering impressive performance. Nonetheless, these existing transformer-based methods often neglect an essential aspect: the incorporation of uncertainty into the predicted series, which holds significant value in decision-making. In this paper, we introduce a **T**ransformer-**M**odulated **D**iffusion **M**odel (**TMDM**), uniting conditional diffusion generative process with transformers into a unified framework to enable precise distribution forecasting for MTS. TMDM harnesses the power of transformers to extract essential insights from historical time series data. This information is then utilized as prior knowledge, capturing covariate-dependence in both the forward and reverse processes within the diffusion model. Furthermore, we seamlessly integrate well-designed transformer-based forecasting methods into TMDM to enhance its overall performance. Additionally, we introduce two novel metrics for evaluating uncertainty estimation performance. Through extensive experiments on six datasets using four evaluation metrics, we establish the effectiveness of TMDM in probabilistic MTS forecasting.

## 1 Introduction

Time series forecasting plays a pivotal role in both the business and scientific domains of machine learning, serving as a vital tool for supporting decision-making in a wide array of downstream applications. These applications encompass, but are not limited to, financial pricing analysis (Kim, 2003), transportation planning (Sapankevych & Sankar, 2009), and weather pattern prediction (Chatfield, 2000), among various other fields (Rasul et al., 2022). The primary objective of time series forecasting is to predict the response variable $\boldsymbol{y}_{0:M} \in \mathbb{R}^{d \times M}$ based on a historical time series dataset represented as $\boldsymbol{x}_{0:N} \in \mathbb{R}^{d \times N}$. This prediction process is characterized by the function $f(\boldsymbol{x}_{0:N}) \in \mathbb{R}^{d \times M}$, where $f$ is a deterministic function that transforms the historical time series $\boldsymbol{x}_{0:N}$ into the future time series $\boldsymbol{y}_{0:M}$.

Existing time series forecasting methods commonly adopt an additive noise model to represent the future time series $\boldsymbol{y}_{0:M}$ as the following: $\boldsymbol{y}_{0:M} = f(\boldsymbol{x}_{0:N}) + \boldsymbol{n}_0$, where $\boldsymbol{n}_0$ follows a normal distribution $\mathcal{N}(0, \boldsymbol{\sigma}^2)$. Consequently, we can calculate the expected value of $\boldsymbol{y}_{0:M}$ given $\boldsymbol{x}_{0:N}$ as $\mathbb{E}[\boldsymbol{y}_{0:M}|\boldsymbol{x}_{0:N}] = f(\boldsymbol{x}_{0:N})$. Classical time series forecasting methods (Liu et al., 2022; Wang et al., 2022; Zhou et al., 2021) that rely on this additive noise model typically provide univariate forecasts by accurately estimating the conditional mean $\mathbb{E}[\boldsymbol{y}_{0:M}|\boldsymbol{x}_{0:N}]$. These models have shown significant advancements in recent years, particularly with the adoption of transformer-based architectures. Transformers leverage self-attention mechanisms and designs tailored for handling time series characteristics effectively. This enhancement empowers transformers to excel in modeling long-term

---

[*]Corresponding author.

dependencies within sequential data (Wu et al., 2021), enabling the development of more potent, large-scale models (Kenton & Toutanova, 2019).

However, the aforementioned methods pay less attention to whether the noise distribution can accurately capture the uncertainty of $\boldsymbol{y}_{0:M}$ given $\boldsymbol{x}_{0:N}$. In time series forecasting, modeling uncertainty holds paramount importance as it directly affects our ability to assess the reliability of predictions for downstream applications (Rasul et al., 2021b). This uncertainty significantly impacts decision-making accuracy. For instance, if a point estimation model predicts that the conditional mean $\mathbb{E}[\boldsymbol{y}_{0:M}|\boldsymbol{x}_{0:N}]$ for tomorrow's temperature is 12 °C, individuals still face a difficult decision regarding whether to cultivate plants today, as the morning temperature might plummet to just 4 °C, jeopardizing the plants' survival. Such models overlook the risks associated with uncertainty, which can be particularly crucial in certain contexts (Kim, 2003; Sapankevych & Sankar, 2009). As another example, if we assign the predicted temperature for tomorrow as a Gaussian distribution, the corresponding uncertainty, represented by $\mathcal{N}(12, 2^2)$ or $\mathcal{N}(12, 8^2)$, could directly influence decision-making processes. The primary objective of this paper is to recover the full distribution of future time series $\boldsymbol{y}_{0:M}$, conditioned on the representation captured by existing well-designed transformer-based models. To achieve this goal,we introduce a novel framework called the **T**ransformer-**M**odulated **D**iffusion **M**odel (**TMDM**), which unifies the conditional diffusion generative process (Ho et al., 2020; Sohl-Dickstein et al., 2015; Song et al., 2020) with transformers, facilitating accurate distribution forecasting for time series.

Recently, diffusion-based generative models have garnered significant attention due to their capacity to generate high-dimensional data and provide training stability (Han et al., 2022). These models can be viewed from various perspectives, including score matching (Hyvärinen & Dayan, 2005; Vincent, 2011) and Langevin dynamics (Neal et al., 2011; Welling & Teh, 2011). However, there has been a recent development in our understanding of these models through the lens of diffusion probabilistic models (Graikos et al., 2022). These models initially employ a forward process to transform data into noise and subsequently use a reverse process to regenerate the data from the noise (Ho et al., 2020).

Current time-series diffusion models (Rasul et al., 2021a; Tashiro et al., 2021; Alcaraz & Strodthoff, 2022; Shen & Kwok, 2023) primarily concentrate on crafting effective conditional embeddings to be fed into the denoising network, which in turn guides the reverse process within the diffusion model. For instance, TimeGrad (Rasul et al., 2021a) employs the hidden state from an RNN as the conditional embedding, while TimeDiff (Shen & Kwok, 2023) constructs this embedding based on two features explicitly designed for time series data. In contrast to prevailing approaches that solely utilize conditional embeddings during the reverse process, TMDM employs conditional information as prior knowledge for both the forward and reverse processes. We believe this approach to be a more efficient means of leveraging the representations captured by existing transformer-based time-series models (Liu et al., 2022; Wang et al., 2022) as conditions, given their proficiency in estimating the conditional mean $\mathbb{E}[\boldsymbol{y}_{0:M} \mid \boldsymbol{x}_{0:N}]$. Empowered by this potent prior knowledge, TMDM is geared toward capturing the uncertainty of future time series $\boldsymbol{y}_{0:M}$, ultimately providing a comprehensive estimate of the entire distribution.

We summarize our contributions as follows: (1) In the realm of probabilistic multivariate time series forecasting, we introduce TMDM, a transformer-based diffusion generative framework. TMDM harnesses the representations captured by well-designed transformer-based time series models as priors. We consider the covariate-dependence across both the forward and reverse processes within the diffusion model, resulting in a highly accurate distribution estimation for future time series. (2) TMDM integrates diffusion and transformer-based models within a cohesive Bayesian framework, employing a hybrid optimization strategy, it serves as a plug-and-play framework, seamlessly compatible with existing well-designed transformer-based forecasting models, leveraging their strong capability to estimate the conditional mean of time series, facilitating the estimation of complete distributions. (3) In our experimental evaluation, we explore the application of Prediction Interval Coverage Probability (PICP) (Yao et al., 2019) and Quantile Interval Coverage Error (QICE) (Han et al., 2022) as metrics in the probabilistic multivariate time series forecasting task. These metrics provide valuable insights into assessing the uncertainty estimation abilities of probabilistic multivariate time series forecasting models. Our study demonstrates TMDM's outstanding performance in four distribution metrics across six real-world datasets, emphasizing its effectiveness in probabilistic MTS forecasting.

## 2 BACKGROUND

### 2.1 DIFFUSION MODEL

Diffusion probabilistic models (Sohl-Dickstein et al., 2015) take the form $p_\theta(\boldsymbol{y}_{0:M}^0) := \int p_\theta(\boldsymbol{y}_{0:M}^{0:T})d\boldsymbol{y}_{0:M}^{1:T}$, where $\boldsymbol{y}_{0:M}^1, ..., \boldsymbol{y}_{0:M}^T$ represent latent variables (Ho et al., 2020). One well-known diffusion model is the denoising diffusion probabilistic model (DDPM) (Ho et al., 2020), which consists of two processes: the forward (diffusion) process and the reverse process. Following the Markov chain, the forward process gradually adds noise, transforming an input vector $\boldsymbol{y}_{0:M}^0$ into a Gaussian noise vector $\boldsymbol{y}_{0:M}^T$ over $T$ steps:

$$q(\boldsymbol{y}_{0:M}^{1:T} \mid \boldsymbol{y}_{0:M}^0) := \prod_{t=1}^T q(\boldsymbol{y}_{0:M}^t \mid \boldsymbol{y}_{0:M}^{t-1}), \quad q(\boldsymbol{y}_{0:M}^t \mid \boldsymbol{y}_{0:M}^{t-1}) := \mathcal{N}(\sqrt{1-\beta^t}\boldsymbol{y}_{0:M}^{t-1}, \beta^t\boldsymbol{I}) \tag{1}$$

where $\beta^t$ represents a small positive constant denoting the noise level. In practical applications, we directly sample $\boldsymbol{y}_{0:M}^t$ from $\boldsymbol{y}_{0:M}^0$ as the following: $q(\boldsymbol{y}_{0:M}^t \mid \boldsymbol{y}_{0:M}^0) = \mathcal{N}(\sqrt{\alpha^t}\boldsymbol{y}_{0:M}^0, (1-\alpha^t)\boldsymbol{I})$, where $\bar{\alpha}^t := 1 - \beta^t$ and $\alpha^t := \prod_{t=1}^T \bar{\alpha}^t$. The reverse process involves denoising $\boldsymbol{y}_{0:M}^t$ back to $\boldsymbol{y}_{0:M}^0$ and is defined as a Markov chain with a learned Gaussian transition:

$$p_\theta(\boldsymbol{y}_{0:M}^{0:T}) := p(\boldsymbol{y}_{0:M}^T)\prod_{t=1}^T p_\theta(\boldsymbol{y}_{0:M}^{t-1} \mid \boldsymbol{y}_{0:M}^t), \quad p_\theta(\boldsymbol{y}_{0:M}^{t-1} \mid \boldsymbol{y}_{0:M}^t) := \mathcal{N}(\boldsymbol{\mu}_\theta(\boldsymbol{y}_{0:M}^t, t), \boldsymbol{\sigma}_\theta(\boldsymbol{y}_{0:M}^t, t)) \tag{2}$$

In DDPM (Ho et al., 2020), the parameterization of $p_\theta(\boldsymbol{y}_{0:M}^{t-1} \mid \boldsymbol{y}_{0:M}^t)$ is defined as:

$$\boldsymbol{\mu}_\theta(\boldsymbol{y}_{0:M}^t, t) = \frac{1}{\alpha^t}(\boldsymbol{y}_{0:M}^t - \frac{\beta^t}{\sqrt{1-\alpha^t}}\boldsymbol{\epsilon}_\theta(\boldsymbol{y}_{0:M}^t, t)),$$
$$\boldsymbol{\sigma}_\theta(\boldsymbol{y}_{0:M}^t, t) = (\bar{\beta}^t)^{1/2}, \text{ if } t = 1: \bar{\beta}^t = \beta^1, \quad \text{else}: \bar{\beta}^t = \frac{1-\alpha^{t-1}}{1-\alpha^t}\beta^t \tag{3}$$

where the $\boldsymbol{\epsilon}_\theta$ is denoising function and which can be trained by solving the following optimization problem:

$$\min_\theta \mathcal{L}(\theta) := \mathbb{E}_{\boldsymbol{y}_{0:M}^0 \sim q(\boldsymbol{y}_{0:M}^0), \boldsymbol{\epsilon} \sim \mathcal{N}(\boldsymbol{0}, \boldsymbol{I}), t} \|\boldsymbol{\epsilon} - \boldsymbol{\epsilon}_\theta(\boldsymbol{y}_{0:M}^t, t))\|^2 \tag{4}$$

Using the trained denoising function $\boldsymbol{\epsilon}_\theta$, we can generate samples step by step from $\mathcal{N}(\boldsymbol{0}, \boldsymbol{I})$ randomly. However, in the context of time series forecasting, the objective is to generate the future time series $\boldsymbol{y}_{0:M}$ conditioned on the historical time series $\boldsymbol{x}_{0:N}$. Several studies (Rasul et al., 2021a; Tashiro et al., 2021; Alcaraz & Strodthoff, 2022; Shen & Kwok, 2023) have explored adapting diffusion models for this task by injecting historical conditional information into the reverse process to guide the generative process.

### 2.2 PROBABILISTIC MULTIVARIATE TIME SERIES FORECASTING

Given an observed history MTS $\boldsymbol{x}_{0:N} = \{x_1, x_2, ..., x_N \mid x_t \in \mathbb{R}^d\}$, probabilistic multivariate time series forecasting tackles the problem of estimating the distribution of the subsequent future time series $\boldsymbol{y}_{0:M} = \{p(y_1), p(y_2), ..., p(y_M) \mid y_t \in \mathbb{R}^d\}$. However, it's essential to note that the exact distribution of $p(y_t)$ is computationally intractable, prompting the development of various methods for its approximation. From the perspective of diffusion generation (Sohl-Dickstein et al., 2015), we establish a Markov chain (Geyer, 1992) with learned Gaussian transitions, initiating from $p(\boldsymbol{y}_{0:M}^T) = \mathcal{N}(\boldsymbol{0}, \boldsymbol{I})$, to estimate the distribution of $\boldsymbol{y}_{0:M}$.

$$p_\theta(\boldsymbol{y}_{0:M}^{0:T} \mid \boldsymbol{x}_{0:N}) := p(\boldsymbol{y}_{0:M}^T)\prod_{t=1}^T p_\theta(\boldsymbol{y}_{0:M}^{t-1} \mid \boldsymbol{y}_{0:M}^t, \boldsymbol{x}_{0:N})$$
$$p_\theta(\boldsymbol{y}_{0:M}^{t-1} \mid \boldsymbol{y}_{0:M}^t, \boldsymbol{x}_{0:N}) := \mathcal{N}(\boldsymbol{\mu}_\theta(\boldsymbol{y}_{0:M}^t, \boldsymbol{x}_{0:N}, t), \boldsymbol{\sigma}_\theta(\boldsymbol{y}_{0:M}^t, \boldsymbol{x}_{0:N}, t)) \tag{5}$$

An essential premise of probabilistic multivariate time series forecasting is the continuity and inter-relation between the observed history $\boldsymbol{x}_{0:N}$ and the future time series $\boldsymbol{y}_{0:M}$. However, two significant challenges arise: firstly, how to extract valuable time series information from $\boldsymbol{x}_{0:N}$ (Shen & Kwok, 2023), and secondly, how to effectively employ this information to guide the generative process. Existing transformer-based time series forecasting models (Liu et al., 2022; Wu et al., 2021; Wang et al., 2022; Zhou et al., 2021) tend to overlook the estimation of time series uncertainty.However, they still have specially designed structures to capture information in the time series, which can be used as conditional information directly. Regarding the second challenge, recent models (Rasul et al., 2021a; Alcaraz & Strodthoff, 2022) attempt to inject conditional embeddings into the denoising network during the reverse process. In contrast to these methods, our proposed model,

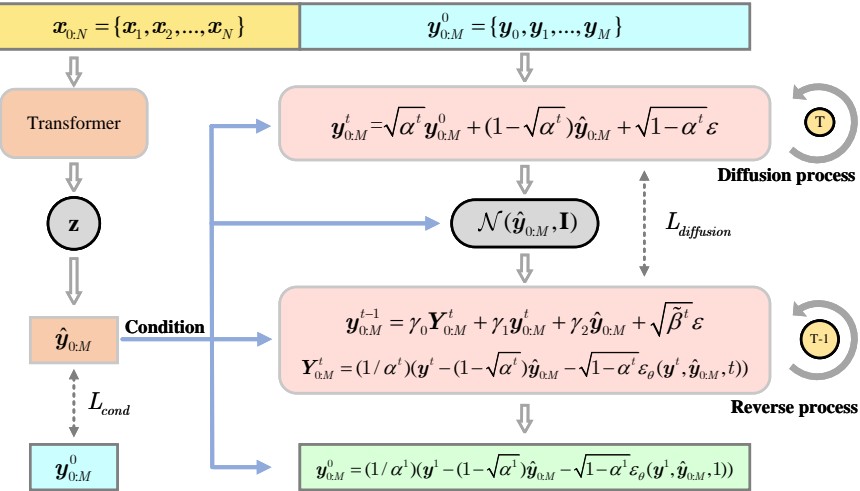

Figure 1: An illustration of the proposed TMDM. The left part is the condition generative model, containing an existing well-designed transformer, used to generate the condition $\hat{y}_{0:M}$. The right part is the proposed conditional diffusion-based time series generative model that utilizes $\hat{y}_{0:M}$ as the prior and introduces covariate-dependence into both the forward and reverse diffusion chains.

TMDM, employs conditional information as prior knowledge for both the forward and reverse processes within the diffusion model. By integrating conditional information into the forward process, TMDM can consider a richer set of conditional information during the denoising process. This enhancement enables TMDM to better learn the inherent time series properties between $x_{0:N}$ (captured by the condition) and $y_{0:M}$ (the generated target).

### 2.3 PICP AND QICE FOR ASSESSING UNCERTAINTY ESTIMATION

To enhance the assessment of uncertainty estimation capabilities in probabilistic multivariate time series forecasting tasks, we introduce two novel metrics: Prediction Interval Coverage Probability (PICP) (Yao et al., 2019) and Quantile Interval Coverage Error (QICE) (Han et al., 2022). The computation of PICP is as the following:

$$\text{PICP} := \frac{1}{N} \sum_{n=1}^{N} \mathbb{1}_{y_n \geq \hat{y}_n^{\text{low}}} \cdot \mathbb{1}_{y_n \leq \hat{y}_n^{\text{high}}} \tag{6}$$

where $\hat{y}_n^{\text{low}}$ and $\hat{y}_n^{\text{high}}$ represent the low and high percentiles, respectively, of our choice for the predicted $y_{0:M}$ outputs given the same $x_{0:N}$ input. In cases where the learned distribution accurately represents the true distribution, this measurement should closely align with the difference between the selected low and high percentiles (Han et al., 2022). QICE can be viewed as an extension of PICP with higher granularity and without any uncovered quantile ranges. Its computation is as the following:

$$\text{QICE} := \frac{1}{M} \sum_{m=1}^{M} \left| r_m - \frac{1}{M} \right|, \text{ where } r_m = \frac{1}{N} \sum_{n=1}^{N} \mathbb{1}_{y_n \geq \hat{y}_n^{\text{low}}{}_m} \cdot \mathbb{1}_{y_n \leq \hat{y}_n^{\text{high}}{}_m}. \tag{7}$$

With a sufficient number of $y_{0:M}$ samples, the first step involves dividing them into $M$ quantile intervals (QIs), each with approximately equal sizes. Subsequently, quantile values corresponding to each QI boundary are determined. In contrast to PICP, QICE offers a more detailed evaluation. In cases where fewer true instances fall within one QI, another QI may capture more instances, potentially leading to increased absolute errors in both QIs. Additionally, we utilize the Continuous Ranked Probability Score (CRPS) (Matheson & Winkler, 1976; Gneiting & Raftery, 2007) and CRPSsum for evaluation on each dimension of the time series. CRPSsum represents the CRPS computed for the sum of all time series dimensions.

### 3 PROPOSED METHOD

In this section, we present TMDM, a novel framework that combines the diffusion generative process (Ho et al., 2020; Sohl-Dickstein et al., 2015) with well-designed transformer structures (Liu

et al., 2022; Wang et al., 2022). These transformer models excel at accurately estimating the conditional mean $\mathbb{E}[\boldsymbol{y}_{0:M} \mid \boldsymbol{x}_{0:N}]$, while TMDM extends this capability to recover the full distribution of the future time series $\boldsymbol{y}_{0:M}$. As depicted in Fig. 1, TMDM consists of two main components: a transformer-powered conditional distribution learning model (condition generative model) and a conditional diffusion-based time series generative model. These two models are integrated into a unified Bayesian framework, leveraging a hybrid optimization approach. From a conceptual standpoint, TMDM can be viewed as a Bayesian generative model (Tran et al., 2019), where the generative process can be expressed as:

$$p(\boldsymbol{y}_{0:M}^0) = \int_{\boldsymbol{y}_{0:M}^{1:T}} \int_{\boldsymbol{z}} p(\boldsymbol{y}_{0:M}^T \mid \hat{\boldsymbol{y}}_{0:M}) \prod_{t=1}^T p(\boldsymbol{y}_{0:M}^{t-1} \mid \boldsymbol{y}_{0:M}^t, \hat{\boldsymbol{y}}_{0:M}) p(\hat{\boldsymbol{y}}_{0:M} \mid \boldsymbol{z}) p(\boldsymbol{z}) d\boldsymbol{z} d\boldsymbol{y}_{0:M}^{1:T} \qquad (8)$$

In this paper, we leverage well-designed transformers, including the Non-stationary transformer (Liu et al., 2022), Autoformer (Wu et al., 2021), and Informer (Zhou et al., 2021), to capture the information embedded within the historical time series $\boldsymbol{x}_{0:M}$. We utilize this information to model a latent variable $\boldsymbol{z}$, which in turn generates a conditional representation $\hat{\boldsymbol{y}}_{0:M}$. This representation serves as a condition for the subsequent forward and reverse processes.

## 3.1 LEARNING TRANSFORMER POWERED CONDITIONS

Existing time-series diffusion models (Rasul et al., 2021a; Tashiro et al., 2021; Alcaraz & Strodthoff, 2022; Shen & Kwok, 2023) have primarily focused on designing effective conditional embeddings to guide the reverse process. In contrast, our approach advocates the utilization of representations captured by well-established transformer-based time series models. This shift offers several distinct advantages. Firstly, significant advancements have been made in point estimation time series forecasting tasks in recent years. Extensive research into time series properties has resulted in the proposal of dedicated transformers tailored for this purpose (Liu et al., 2022; Wu et al., 2021; Wang et al., 2022). We contend that employing conditions derived from such transformers is more efficient than relying on self-designed conditioning embeddings. Secondly, these specialized transformers exhibit a strong capability to estimate the conditional mean $\mathbb{E}[\boldsymbol{y}_{0:M}|\boldsymbol{x}_{0:N}]$. By employing this estimated mean as the condition, the diffusion model can more effectively focus on estimating uncertainty, simplifying the generative process. Conversely, using other specially designed conditions, such as future mixup (Shen & Kwok, 2023), may introduce new information but requires the diffusion model to simultaneously estimate both the mean and uncertainty, rendering generation more complex. Finally, TMDM serves as a versatile plug-and-play framework, bridging the gap between point estimates and distribution estimates. If improved transformer structures for point estimation emerge, we can seamlessly integrate these advancements into the distribution estimation domain.

Given the transformer structure $\mathscr{T}(\cdot)$ and the historical time series $\boldsymbol{x}_{0:N}$, we can capture the representation by $\mathscr{T}(\boldsymbol{x}_{0:N})$. This representation serves as the guiding factor for approximating the true posterior distribution of $\boldsymbol{z}$. This process is defined as the following:

$$q(\boldsymbol{z} \mid \mathscr{T}(\boldsymbol{x}_{0:N})) \sim \mathcal{N}\left(\tilde{\boldsymbol{\mu}}_z(\mathscr{T}(\boldsymbol{x}_{0:N})), \tilde{\boldsymbol{\sigma}}_z(\mathscr{T}(\boldsymbol{x}_{0:N}))\right) \qquad (9)$$

Given a well-learned $\boldsymbol{z}$, we can generate the conditional representation $\hat{\boldsymbol{y}}_{0:M}$ as the following:

$$\boldsymbol{z} \sim \mathcal{N}(0, 1) \quad \text{and} \quad \hat{\boldsymbol{y}}_{0:M} \sim \mathcal{N}\left(\boldsymbol{\mu}_z(\boldsymbol{z}), \boldsymbol{\sigma}_z\right) \qquad (10)$$

Here, we model three nonlinearity functions, $\tilde{\boldsymbol{\mu}}_z$, $\tilde{\boldsymbol{\sigma}}_z$, and $\boldsymbol{\mu}_z$, using neural networks. We initialize $\boldsymbol{\sigma}_z$, representing the covariance matrix, to the identity matrix $\boldsymbol{I}$. In this manner, we define a latent variable $\boldsymbol{z}$ to summarize the information captured by well-designed transformers. This latent variable is then used to generate the conditional representation $\hat{\boldsymbol{y}}_{0:M}$ for the subsequent forward and reverse processes in TMDM.

## 3.2 CONDITIONAL DIFFUSION-BASED TIME SERIES GENERATIVE MODEL

Different from vanilla diffusion models that assume the endpoint of the diffusion process, $\boldsymbol{y}_{0:M}^T$, adheres to the standard normal distribution $\mathcal{N}(0, 1)$, we incorporate the conditional representation $\hat{\boldsymbol{y}}_{0:M}$ into $p(\boldsymbol{y}_{0:M}^T)$ to better account for the conditional information in Eq. 10. Drawing inspiration from Han et al. (2022), we model the endpoint of our diffusion process as the following:

$$p(\boldsymbol{y}_{0:M}^T \mid \hat{\boldsymbol{y}}_{0:M}) = \mathcal{N}(\hat{\boldsymbol{y}}_{0:M}, \boldsymbol{I}) \qquad (11)$$

where $\hat{\boldsymbol{y}}_{0:M}$, as defined in Eq. 10, incorporates the information captured by the transformer. In Eq. 11, $\hat{\boldsymbol{y}}_{0:M}$ can be viewed as prior knowledge for estimating the conditional mean $\mathbb{E}[\boldsymbol{y}_{0:M} \mid \boldsymbol{x}_{0:N}]$

| **Algorithm 1** Training | **Algorithm 2** Inference |
|---|---|
| 1: Initialize the parameters; | 1: $\boldsymbol{y}_{0:M}^T \sim \mathcal{N}(\hat{\boldsymbol{y}}_{0:M}, \boldsymbol{I})$ |
| 2: **repeat** | 2: **for** t = T to 1 **do** |
| 3:    Draw $\boldsymbol{y}_{0:M}^0 \sim q(\boldsymbol{y}_{0:M}^0 \mid \boldsymbol{x}_{0:N})$ | 3:    Calculate reparameterize: $\boldsymbol{Y}_{0:M}^t = (1/\alpha^t)(\boldsymbol{y}^t - (1 -$ |
| 4:    Draw $t \sim \text{Uniform}(\{1, 2, ..., T\})$ |     $\sqrt{\alpha^t})\hat{\boldsymbol{y}}_{0:M} - \sqrt{1 - \alpha^t}\varepsilon_\theta(\boldsymbol{y}^t, \hat{\boldsymbol{y}}_{0:M}, t))$ |
| 5:    Draw $\boldsymbol{\epsilon} \sim \mathcal{N}(0, 1)$ | 4:    **if** $t > 1$: draw $\boldsymbol{\epsilon} \sim \mathcal{N}(0, 1)$ |
| 6:    Compute the loss in Eq. 16 | 5:        $\boldsymbol{y}_{0:M}^{t-1} = \gamma_0 \boldsymbol{Y}_{0:M}^t + \gamma_1 \boldsymbol{y}_{0:M}^t + \gamma_2 \hat{\boldsymbol{y}}_{0:M} + \sqrt{\tilde{\beta}^t}\boldsymbol{\epsilon}$ |
| 7:    Take numerical optimization step | 6:    **else:**  $\boldsymbol{y}_{0:M}^{t-1} = \boldsymbol{Y}_{0:M}^t$ |
|     on: $\nabla \mathcal{L}_{\text{ELBO}}$ | 7: **end for** |
| 8: **until** converged | |

based on $\boldsymbol{x}_{0:N}$. With a diffusion schedule $\{\beta^t\}_{t=1:T} \in (0, 1)$, the conditional distributions for the forward process at all other time steps can be defined as:

$$q\left(\boldsymbol{y}_{0:M}^t \mid \boldsymbol{y}_{0:M}^{t-1}, \hat{\boldsymbol{y}}_{0:M}\right) \sim \mathcal{N}\left(\boldsymbol{y}_{0:M}^t \mid \sqrt{1 - \beta^t}\boldsymbol{y}_{0:M}^{t-1} + (1 - \sqrt{1 - \beta^t})\hat{\boldsymbol{y}}_{0:M}, \beta^t \boldsymbol{I}\right) \qquad (12)$$

In practical applications, we sample $\boldsymbol{y}_{0:M}^t$ directly from $\boldsymbol{y}_{0:M}^0$ with an arbitrary timestep $t$:

$$q\left(\boldsymbol{y}_{0:M}^t \mid \boldsymbol{y}_{0:M}^0, \hat{\boldsymbol{y}}_{0:M}\right) \sim \mathcal{N}\left(\boldsymbol{y}_{0:M}^t \mid \sqrt{\alpha^t}\boldsymbol{y}_{0:M}^0 + (1 - \sqrt{\alpha^t})\hat{\boldsymbol{y}}_{0:M}, (1 - \sqrt{\alpha^t})\boldsymbol{I}\right) \qquad (13)$$

Here, we define $\bar{\alpha}^t := 1 - \beta^t$ and $\alpha^t := \prod_{t=1}^T \bar{\alpha}^t$. In Eq. 12 mean term, the diffusion process can be conceptualized as an interpolation between the true data $\boldsymbol{y}_{0:M}^0$ and the conditional representation $\hat{\boldsymbol{y}}_{0:M}$. It commences with the true data $\boldsymbol{y}_{0:M}^0$ and gradually transitions to $\hat{\boldsymbol{y}}_{0:M}$. This approach effectively leverages the reliable conditional mean estimation $\mathbb{E}[\boldsymbol{y}_{0:M}|\boldsymbol{x}_{0:N}]$ by transformers $\mathscr{T}(\cdot)$. In the corresponding reverse process, initiated with $\hat{\boldsymbol{y}}_{0:M}$ containing information capable of accurately estimating $\mathbb{E}[\boldsymbol{y}_{0:M}|\boldsymbol{x}_{0:N}]$, the generative process is significantly simplified. If the provided condition is good enough, the model can then focus exclusively on uncertainty estimation.

Similar to many diffusion models designed for time series (Rasul et al., 2021a; Shen & Kwok, 2023), it is crucial for the reverse process to incorporate the conditional representation $\hat{\boldsymbol{y}}_{0:M}$. Considering the forward process in Eq. 12, the corresponding manageable posterior for the forward process is:

$$q\left(\boldsymbol{y}_{0:M}^{t-1} \mid \boldsymbol{y}_{0:M}^0, \boldsymbol{y}_{0:M}^t, \hat{\boldsymbol{y}}_{0:M}\right) \sim \mathcal{N}\left(\boldsymbol{y}_{0:M}^{t-1} \mid \gamma_0 \boldsymbol{y}_{0:M}^0 + \gamma_1 \boldsymbol{y}_{0:M}^t + \gamma_2 \hat{\boldsymbol{y}}_{0:M}, \tilde{\beta}^t \boldsymbol{I}\right)$$
$$\gamma_0 = \frac{\beta^t \sqrt{\alpha^{t-1}}}{1 - \alpha^t}, \gamma_1 = \frac{(1 - \alpha^{t-1})\sqrt{\bar{\alpha}^t}}{1 - \alpha^t}, \gamma_2 = 1 + \frac{(\sqrt{\alpha^t} - 1)(\sqrt{\bar{\alpha}^t} + \sqrt{\alpha^{t-1}})}{1 - \alpha^t}, \tilde{\beta}^t = \frac{(1 - \alpha^{t-1})}{1 - \alpha^t}\beta^t \qquad (14)$$

The derivation can be find in Appendix D.

### 3.3 HYBRID OPTIMIZATION

In this paper, we integrate the condition generative model and denoising model into a unified optimization objective. The condition generative model incorporates transformer $\mathscr{T}(\cdot)$ structures and networks associated with the latent variable $\boldsymbol{z}$. Within the diffusion model component, a denoising model is trained. As specified in Eq. 8, TMDM's optimization objective is to maximize the evidence lower bound (ELBO) of the log marginal likelihood, formulated as:

$$\log p\left(\boldsymbol{y}_{0:M}^0 \mid \boldsymbol{x}_{0:N}\right) \geq \log \mathbb{E}_{q(\boldsymbol{y}_{0:M}^{1:T}, \boldsymbol{z}|\boldsymbol{y}_{0:M}^0, \hat{\boldsymbol{y}}_{0:M}, \mathscr{T}(\boldsymbol{x}_{0:N}))}\left[\frac{p(\boldsymbol{y}_{0:M}^{0:T}|\hat{\boldsymbol{y}}_{0:M})p(\hat{\boldsymbol{y}}_{0:M}|\boldsymbol{z})p(\boldsymbol{z})}{q(\boldsymbol{y}_{0:M}^{1:T}, \boldsymbol{z}|\boldsymbol{y}_{0:M}^0, \hat{\boldsymbol{y}}_{0:M}, \mathscr{T}(\boldsymbol{x}_{0:N}))}\right]$$

$$= \mathbb{E}_{q(\boldsymbol{y}_{0:M}^{1:T}|\boldsymbol{y}_{0:M}^0, \hat{\boldsymbol{y}}_{0:M})}\left[\log \frac{p(\boldsymbol{y}_{0:M}^{0:T}|\hat{\boldsymbol{y}}_{0:M})}{q(\boldsymbol{y}_{0:M}^{1:T}|\boldsymbol{y}_{0:M}^0, \hat{\boldsymbol{y}}_{0:M})}\right] + \mathbb{E}_{q(\boldsymbol{z}|\mathscr{T}(\boldsymbol{x}_{0:N}))}\left[\log \frac{p(\hat{\boldsymbol{y}}_{0:M}|\boldsymbol{z})p(\boldsymbol{z})}{q(\boldsymbol{z}|\mathscr{T}(\boldsymbol{x}_{0:N}))}\right]$$
$$(15)$$

In Eq. 15, the first term, denoted as $\mathcal{L}_{\text{diffusion}}$, guides the denoising model to predict uncertainty while subtly adjusting the condition generative model to offer a more suitable conditional representation. We view this as an advantage of hybrid optimization. The second term, $\mathcal{L}_{\text{cond}}$, is introduced to maintain the capacity for accurate estimation of the conditional mean $\mathbb{E}[\boldsymbol{y}_{0:M}|\boldsymbol{x}_{0:N}]$ by the condition generative model. It also facilitates the generation of improved conditional representations by leveraging the capabilities of a well-designed transformer. Here, $\mathbf{D}_{KL}(q\|p)$ represents the Kullback–Leibler (KL) divergence from distribution $p$ to distribution $q$. The aforementioned objective

can be expressed as:

$$\mathcal{L}_{\text{ELBO}} = \mathbb{E}_q[-\log p(\boldsymbol{y}_{0:M}^0 \mid \boldsymbol{y}_{0:M}^1, \hat{\boldsymbol{y}}_{0:M})] + \mathbf{D}_{KL}(q(\boldsymbol{y}_{0:M}^T \mid \boldsymbol{y}_{0:M}^0, \hat{\boldsymbol{y}}_{0:M}) \| p(\boldsymbol{y}_{0:M}^T \mid \hat{\boldsymbol{y}}_{0:M}))$$

$$+ \sum_{t=2}^{T} \mathbf{D}_{KL}(q\left(\boldsymbol{y}_{0:M}^{t-1} \mid \boldsymbol{y}_{0:M}^0, \boldsymbol{y}_{0:M}^t, \hat{\boldsymbol{y}}_{0:M}\right) \| p(\boldsymbol{y}_{0:M}^{t-1} \mid \boldsymbol{y}_{0:M}^t, \hat{\boldsymbol{y}}_{0:M})) \qquad (16)$$

$$+ \mathbb{E}_{q(\boldsymbol{z} \mid \mathscr{T}(\boldsymbol{x}_{0:N}))}[-\log p(\hat{\boldsymbol{y}}_{0:M} \mid \boldsymbol{z})] + \mathbf{D}_{KL}(q\left(\boldsymbol{z} \mid \mathscr{T}(\boldsymbol{x}_{0:N})\right) \| p(\boldsymbol{z}))$$

In Eq. 16, the first two rows originate from $\mathcal{L}_{\text{diffusion}}$, while the last rows stem from $\mathcal{L}_{\text{cond}}$. As depicted in Algorithm 1, the model parameters are optimized through stochastic gradient descent in an end-to-end manner. The inference process is outlined in Algorithm 2.

## 4 EXPERIMENTS

### 4.1 EXPERIMENT SETUP

**Dataset:** Six real-world datasets with diverse spatiotemporal dynamics were chosen, comprising Electricity, ILI, ETT, Exchange, Traffic, and Weather. Table 1 presents basic statistical information about these datasets. Further details can be found in Appendix A.

Table 1: Summary of dataset statistics.

| Dataset | Dimension | Freq. | Time steps | Pred. steps |
|---|---|---|---|---|
| Exchange | 8 | 1 Day | 7,588 | 192 |
| ILI | 7 | 1 Week | 966 | 36 |
| ETTm2 | 7 | 15 Min | 69,680 | 192 |
| Electricity | 321 | 1 Hour | 26,304 | 192 |
| Traffic | 862 | 1 Hour | 17,544 | 192 |
| Weather | 21 | 10 Min | 52,695 | 192 |

**Implementation details:** In our experiments, we set the number of timesteps as $T = 1000$ and employed a linear noise schedule with $\beta^1 = 10^{-4}$ and $\beta^T = 0.02$, consistent with the setup in Ho et al. (2020). For the PICP, we selected the 2.5th and 97.5th percentiles. Therefore, an ideal PICP value for the learned model should be 95%. We employed 100 samples to approximate the estimated distribution, and all experiments were repeated 10 times, with mean and standard deviation recorded. More details can be find in Appendix B.

### 4.2 MAIN RESULT

#### 4.2.1 BASELINES

We extensively compare our model with 14 baselines with different experiment settings. Including diffusion-based time series models: TimeGrad (Rasul et al., 2021a), CSDI (Tashiro et al., 2021), SSSD (Alcaraz & Strodthoff, 2022), D³VAE (Li et al., 2022) and TimeDiff (Shen & Kwok, 2023); Transformer-based models: Transformer-MAF (Rasul et al., 2021b), Transformer (Vaswani et al., 2017), Informer (Zhou et al., 2021), Autoformer (Wu et al., 2021) and NSformer (Liu et al., 2022); VAE-based models: VAE (Higgins et al., 2016), cST-ML (Zhang et al., 2020) and DAC-ML (Zhang et al., 2021); and one additional well-designed method: GP-Copula (Salinas et al., 2019).

#### 4.2.2 QUALITATIVE ANALYSIS

To emphasize our distribution estimation capabilities, we present the predicted median and visualize the 50% and 90% distribution intervals in Fig. 2. We compare TMDM with three other models: *TMDM-min*: A simplified version of TMDM that employs a basic transformer in the condition generative model. *TimeDiff*: A recent time series prediction model that operates in a non-autoregressive setting. However, it was primarily designed for point-to-point forecasting tasks, which may not prioritize probabilistic forecasting. *TimeGrad*: A well-known diffusion-based autoregressive model.

Overall, *TMDM* demonstrates superior distribution estimation performance compared to the other three models. While *TMDM-min* exhibits worse mean and uncertainty estimates than *TMDM*, we attribute this to the use of a different transformer in the condition generative model. The NSformer employed in TMDM is more powerful for mean estimation, facilitating better overall distribution estimation. *TimeDiff*, designed for point-to-point forecasting tasks, generates distribution intervals with varying width. Consequently, sample points generated farther away from the true value are sparse. In challenging scenarios (columns 3, 4, and 5), the 50% distribution intervals abruptly expand due to the absence of points in the middle section. This highlights the limitation of point-to-point forecasting in capturing real multivariate time series data, making it less practical in real-world applications. *TimeGrad*, which relies on an RNN to capture timing information, exhibits

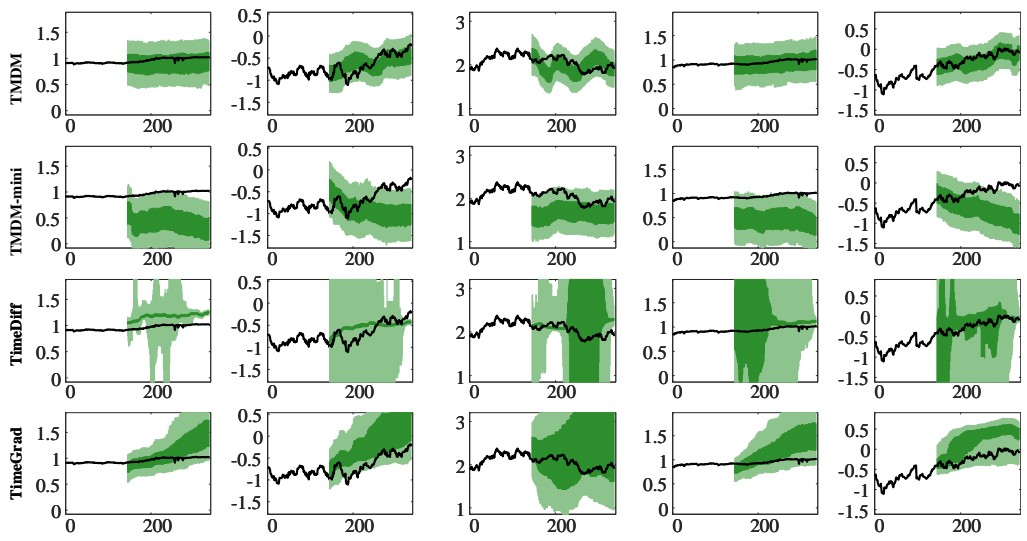

Figure 2: Comparison of prediction intervals for the Exchange dataset. We display the predicted median and visualize the 50% and 90% distribution intervals, the black line representing the test set ground-truth.

Table 2: Performance comparisons on six real-world datasets in terms of QICE and CRPS. The best results are boldfaced.

| Dataset | Exchange | | ILI | | ETTm2 | | Electricity | | Traffic | | Weather | |
|---|---|---|---|---|---|---|---|---|---|---|---|---|
| Method | QICE | CRPS | QICE | CRPS | QICE | CRPS | QICE | CRPS | QICE | CRPS | QICE | CRPS |
| VAE | 8.28±0.590 | 1.02±0.112 | 9.13±0.492 | 2.41±0.185 | 8.99±0.121 | 0.79±0.057 | 7.04±0.080 | 0.51±0.162 | 5.37±0.197 | 0.67±0.139 | 9.07±0.669 | 0.47±0.155 |
| cST-ML | 7.94±0.786 | 0.94±0.264 | 9.02±0.570 | 1.94±0.198 | 7.29±0.161 | 0.64±0.122 | 5.99±0.167 | 0.47±0.181 | 5.24±0.134 | 0.60±0.115 | 8.29±0.760 | 0.51±0.120 |
| DAC-ML | 7.36±0.709 | 0.85±0.183 | 8.71±0.586 | 1.23±0.114 | 6.60±0.191 | 0.59±0.109 | 5.76±0.153 | 0.43±0.147 | 4.31±0.165 | 0.50±0.088 | 7.91±0.823 | 0.46±0.115 |
| GP-Copula | 7.71±0.763 | 0.88±0.201 | 8.95±0.756 | 1.43±0.305 | 6.86±0.080 | 0.59±0.187 | 5.85±0.192 | 0.86±0.160 | 4.82±0.165 | 0.53±0.037 | 8.10±0.545 | 0.84±0.275 |
| Transformer-MAF | 6.85±0.818 | 0.71±0.202 | 8.19±0.669 | 0.98±0.149 | 5.60±0.110 | 0.66±0.123 | 5.66±0.114 | 0.46±0.132 | 3.95±0.086 | 0.45±0.125 | 7.40±0.777 | 0.45±0.152 |
| TimeGrad | 5.32±0.826 | 0.66±0.192 | 7.86±1.126 | 0.92±0.142 | 5.37±0.187 | 0.54±0.184 | 5.34±0.197 | 0.40±0.176 | 3.80±0.150 | 0.39±0.166 | 7.36±0.871 | 0.43±0.116 |
| SSSD | 6.20±0.569 | 0.56±0.174 | 7.60±0.739 | 0.94±0.195 | 4.88±0.166 | 0.57±0.179 | 5.26±0.177 | 0.46±0.266 | 3.88±0.140 | 0.41±0.114 | 7.19±0.674 | 0.44±0.125 |
| CSDI | 5.49±0.606 | 0.45±0.138 | 7.75±0.389 | 1.10±0.144 | 5.07±0.132 | 0.50±0.112 | 4.74±0.175 | 0.42±0.094 | 3.50±0.271 | 0.37±0.121 | 5.14±0.700 | 0.37±0.086 |
| D³VAE | 13.51±0.602 | 0.41±0.069 | 15.82±0.190 | 0.97±0.072 | 13.48±0.111 | 0.44±0.014 | 13.41±.117 | 0.36±0.120 | 12.60±0.113 | 0.31±0.114 | 14.64±0.607 | 0.39±0.026 |
| TimeDiff | 13.34±0.610 | 0.38±0.036 | 15.50±0.098 | 1.08±0.063 | 14.22±0.164 | 0.40±0.003 | 12.74±0.105 | 0.38±0.097 | 13.53±0.175 | 0.28±0.115 | 13.18±0.788 | **0.33**±0.016 |
| ours | **4.38**±0.417 | **0.32**±0.016 | **6.74**±0.082 | **0.92**±0.071 | **3.75**±0.138 | **0.37**±0.007 | **3.81**±0.133 | **0.33**±0.085 | **2.36**±0.117 | **0.26**±0.091 | **3.87**±0.681 | 0.36±0.038 |

poor performance when estimating longer time series. For more detailed results, please refer to the Appendix E and F.

### 4.2.3 QUANTITATIVE COMPARISON

**Probabilistic multivariate time series forecasting:** To assess the performance of TMDM in probabilistic multivariate time series forecasting, we applied our model to six datasets alongside several competitive baselines. Four metrics (QICE, PICP, CRPS, and CRPS$_{sum}$) were employed as performance indicators for probabilistic forecasting. Additionally, we used two metrics (MSE and MAE) to evaluate other aspects of model performance. As shown in Table 2, the NSformer integrated with our framework consistently achieved state-of-the-art (SOTA) performance across all benchmark datasets. Notably, when compared to previous SOTA results, TMDM achieved a remarkable **17%** reduction in QICE (from 5.32 to 4.38) for the Exchange dataset, **11%** reduction (from 7.6 to 6.74) for ILI, **23%** reduction (from 4.88 to 3.75) for ETTm2, **27%** reduction (from 5.26 to 3.81) for Electricity, **32%** reduction (from 3.5 to 2.36) for Traffic, and **24%** reduction (from 5.14 to 3.87) for Weather. It is worth noting that models like D³VAE (Li et al., 2022) and TimeDiff (Shen & Kwok, 2023) were originally designed for point-to-point forecasting tasks, where probabilistic forecasting was not their primary focus. Consequently, their performance on QICE and PICP was suboptimal. However, they still demonstrated competitive performance in CRPS, CRPS$_{sum}$ (see Appendix C), MSE, and MAE (refer to Table 3). As CRPS may not effectively evaluate the quality of distribution range, this underscores the importance of introducing new metrics (QICE and PICP) in the evaluation of probabilistic multivariate time series forecasting models. For additional results, please refer to the Appendix C.

Table 3: Performance comparisons on six real-world datasets regarding MSE and MAE. The best results are boldfaced.

| Dataet | Exchange | | ILI | | ETTm2 | | Electricity | | Traffic | | Weather | |
|---|---|---|---|---|---|---|---|---|---|---|---|---|
| Method | MSE | MAE | MSE | MAE | MSE | MAE | MSE | MAE | MSE | MAE | MSE | MAE |
| TimeGrad | 2.43±0.229 | 0.90±0.232 | 2.65±0.164 | 1.15±0.172 | 1.36±0.133 | 0.74±0.123 | 0.69±0.188 | 0.74±0.109 | 0.96±0.104 | 0.81±0.141 | 0.90±0.139 | 0.57±0.136 |
| CSDI | 1.67±0.162 | 0.75±0.058 | 2.54±0.098 | 1.21±0.128 | 1.28±0.074 | 0.67±0.064 | 0.56±0.212 | 0.81±0.150 | 0.94±0.093 | 0.68±0.176 | 0.86±0.073 | 0.56±0.096 |
| SSSD | 0.90±0.171 | 0.86±0.127 | 2.52±0.118 | 1.08±0.131 | 0.97±0.043 | 0.56±0.060 | 0.47±0.129 | 0.60±0.207 | 0.81±0.084 | 0.50±0.128 | 0.67±0.159 | 0.49±0.106 |
| D³VAE | 0.76±0.118 | 0.62±0.108 | 2.44±0.115 | 1.11±0.127 | 0.79±0.038 | 0.46±0.047 | 0.33±0.194 | 0.49±0.119 | 0.79±0.122 | 0.43±0.126 | 0.43±0.139 | 0.34±0.131 |
| TimeDiff | 0.48±0.095 | 0.43±0.109 | 2.46±0.148 | 1.09±0.064 | 0.41±0.014 | 0.42±0.013 | 0.27±0.024 | 0.32±0.131 | 0.68±0.113 | 0.47±0.052 | 0.36±0.146 | 0.37±0.052 |
| ours | **0.26±0.019** | **0.37±0.015** | **1.99±0.085** | **0.85±0.026** | **0.27±0.023** | **0.35±0.015** | **0.19±0.007** | **0.27±0.008** | **0.60±0.008** | **0.35±0.009** | **0.28±0.095** | **0.25±0.103** |

**Ablation study:** To assess the impact of each component within our proposed framework, we conducted a comparative analysis of prediction results across three datasets using five models. Presented in Table 4, *MLP-cond* and *Autoformer-cond* serve as baseline models, employing a simple MLP or Autoformer in the condition generative model of TMDM. $\mathcal{N}(\mathbf{0}, \mathbf{I})$-*Prior* leverages NSformer to generate

Table 4: QICE and CRPS scores for different variants of the TMDM

| Dataset | Exchange | | ETTm2 | | Wether | |
|---|---|---|---|---|---|---|
| Metric | QICE | CRPS | QICE | CRPS | QICE | CRPS |
| MLP-cond | 7.86 | 0.64 | 9.61 | 1.14 | 5.06 | 0.50 |
| Autoformer-cond | 6.77 | 0.40 | 4.76 | 0.56 | 4.55 | 0.46 |
| $\mathcal{N}(\mathbf{0}, \mathbf{I})$-Prior | 5.44 | 0.46 | 5.48 | 0.57 | 4.23 | 0.44 |
| No-hybrid | 5.96 | 0.38 | 4.47 | 0.46 | 4.18 | 0.41 |
| TMDM | **4.38** | **0.32** | **3.75** | **0.37** | **3.87** | **0.36** |

conditional embeddings while assuming $\boldsymbol{y}_{0:M}^T \sim \mathcal{N}(\mathbf{0}, \mathbf{I})$ as a prior. When comparing our proposed model, *TMDM*, with *MLP-cond* and *Autoformer-cond*, we observed a substantial improvement, achieving an average QICE reduction of $42\%$ and $23\%$ respectively. This emphasizes the effectiveness of utilizing representations captured by existing well-designed transformer-based models as conditions. Furthermore, it demonstrates our capability to seamlessly integrate advancements in transformer structures for point estimation into the domain of distribution estimation. Comparing $\mathcal{N}(\mathbf{0}, \mathbf{I})$-*Prior* with *TMDM*, we noted an average $19\%$ reduction in QICE, highlighting the advantage of considering condition information as a prior for both the forward and reverse processes. Finally, comparing *No-hybrid* with *TMDM*, we observed a significant $16\%$ reduction in QICE, underscoring the effectiveness of our proposed hybrid optimization method.

Table 5: Performance promotion by applying the proposed framework to transformer and its variants.

| Dataet | Exchange | | ILI (36) | | ETTm2 | | Electricity | | Traffic | | Weather | |
|---|---|---|---|---|---|---|---|---|---|---|---|---|
| Method | MSE | MAE | MSE | MAE | MSE | MAE | MSE | MAE | MSE | MAE | MSE | MAE |
| Transformer | 1.20±0.129 | 0.84±0.041 | 4.93±0.277 | 1.48±0.082 | 4.76±1.214 | 1.76±0.256 | **0.26±0.016** | 0.36±0.015 | 0.67±0.014 | 0.36±0.016 | 0.57±0.045 | 0.53±0.024 |
| + Ours | **1.16±0.023** | **0.82±0.015** | **4.48±0.180** | **1.41±0.039** | **1.18±0.186** | **0.86±0.065** | 0.27±0.009 | **0.33±0.009** | **0.62±0.009** | **0.33±0.010** | **0.55±0.038** | **0.49±0.015** |
| Informer | 1.31±0.154 | 0.85±0.021 | 5.33±0.177 | 1.59±0.078 | 5.74±0.475 | 1.99±0.181 | 0.35±0.012 | 0.43±0.020 | 0.75±0.017 | 0.42±0.019 | 0.48±0.078 | 0.47±0.045 |
| + Ours | **1.12±0.085** | **0.83±0.014** | **4.82±0.089** | **1.49±0.041** | **1.25±0.264** | **0.90±0.096** | **0.34±0.010** | **0.40±0.013** | **0.69±0.007** | **0.40±0.011** | **0.41±0.044** | **0.45±0.027** |
| Autoformer | 0.44±0.146 | 0.48±0.082 | 3.26±0.180 | 1.25±0.066 | 0.47±0.032 | 0.46±0.021 | 0.22±0.019 | 0.33±0.015 | 0.65±0.048 | 0.41±0.028 | 0.31±0.018 | 0.37±0.025 |
| + Ours | **0.43±0.027** | **0.47±0.016** | **3.03±0.101** | **1.18±0.045** | **0.33±0.028** | **0.40±0.018** | **0.20±0.009** | **0.29±0.010** | **0.63±0.022** | **0.38±0.014** | **0.31±0.011** | **0.34±0.019** |
| NSformer | **0.25±0.088** | **0.36±0.091** | **1.93±0.157** | 0.87±0.058 | 0.53±0.042 | 0.48±0.016 | **0.18±0.012** | 0.28±0.012 | 0.62±0.015 | **0.34±0.013** | **0.25±0.016** | 0.29±0.020 |
| + Ours | 0.26±0.019 | 0.37±0.015 | 1.99±0.085 | **0.85±0.026** | **0.27±0.023** | **0.35±0.015** | 0.19±0.007 | **0.27±0.008** | **0.60±0.008** | 0.35±0.009 | 0.26±0.010 | **0.27±0.009** |

**Framework generality:** Diffusion models have gained widespread attention owing to their capacity to generate high-dimensional data and their training stability (Han et al., 2022). In the context of point-to-point forecasting tasks, TMDM operates as a versatile framework that enhances training stability when paired with various transformers. We apply our framework to four prominent transformers, showcasing the performance enhancements achieved by each model in Table 5. Our method consistently reduces variance across multiple experiments and enhances the point-to-point forecasting ability of most transformers.

## 5 CONCLUSION

In this paper, we present TMDM, an innovative framework that merges diffusion generative process with existing well-designed transformer models. Our approach leverages the strengths of transformers, particularly their accuracy in estimating conditional means, and extends this capability as priors across both forward and reverse processes within the diffusion model. By employing this estimated mean as the condition, the diffusion model can more effectively focus on estimating uncertainty, simplifying the generative process. TMDM stands out as a versatile plug-and-play framework, effectively closing the gap between point estimates and distribution estimates. It enables seamless integration with advanced transformer models for point estimation, promising even better forecasting accuracy. We introduce two novel evaluation metrics, enriching the toolbox for assessing uncertainty in probabilistic multivariate time series forecasting models. Our comprehensive experiments on six real-world datasets consistently demonstrate TMDM's superior performance, underscoring its effectiveness in enhancing probabilistic prediction quality.

ACKNOWLEDGMENTS

This work was supported in part by the National Natural Science Foundation of China under Grant U21B2006; in part by Shaanxi Youth Innovation Team Project; in part by the Fundamental Research Funds for the Central Universities QTZX23037 and QTZX22160; in part by the 111 Project under Grant B18039; The work of Wenchao Chen acknowledges the support of the stabilization support of National Radar Signal Processing Laboratory under Grant (JKW202X0X) and National Natural Science Foundation of China (NSFC) (6220010437).

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

## A   DATASETS

Six real-world datasets with diverse spatiotemporal dynamics are selected, including: (1) *Electricity*[†] - records the hourly electricity consumption of 321 clients from 2012 to 2014. (2) *ILI*[‡] - collects the ratio of influenza-like illness (ILI) patients versus the total patients in one week, which is reported weekly by the Centers for Disease Control and Prevention of the United States from 2002 to 2021. (3) *ETT*(Zhou et al., 2021) - contains the data collected from electricity transformers, including load and oil temperature that are recorded every 15 minutes between July 2016 and July 2018. (4) *Exchange*(Lai et al., 2018) - records the daily exchange rates of eight different countries ranging from 1990 to 2016. (5) *Traffic*[§] - contains hourly road occupancy rates measured by 862 sensors on San Francisco Bay area freeways from January 2015 to December 2016. (6) *Weather*[¶] - includes meteorological time series with 21 weather indicators collected every 10 minutes from the Weather Station of the Max Planck Biogeochemistry Institute in 2020. Table 1 provides basic statistical information about these datasets.

## B   IMPLEMENTATION DETAILS

In our experiments, we set the number of timesteps as $T = 1000$ and employed a linear noise schedule with $\beta^1 = 10^{-4}$ and $\beta^T = 0.02$, consistent with the setup in Ho et al. (2020). The latent states $z$ were configured to have a dimension of 512 for all datasets. For the diffusion model, we adopted a simplified network architecture compared to prior work Xiao et al. (2021); Zheng (2022). Initially, we replaced the transformer's sinusoidal position embedding with a linear embedding for the timestep. We concatenated $y0 : M^t$ and $\hat{y}0 : M$ and passed the resulting vector through three fully-connected layers, each with an output dimension of 128. We performed a Hadamard product between each output vector and the corresponding timestep embedding, followed by a Softplus non-linearity, before forwarding the resulting vector to the next fully-connected layer. Finally, we applied a fourth fully-connected layer to map the vector to a one-dimensional output for the forward diffusion noise prediction. We utilized the Adam optimizer with a learning rate of 0.0001 and a batch size of 32. All experiments were implemented in PyTorch (Paszke et al., 2019) and conducted on an NVIDIA RTX 3090 24GB GPU. The prediction length can be found in Table 1. The input sequence length for ILI is set to 36, while for other datasets, it is set to 96. For the Prediction Interval Coverage Probability (PICP), we selected the 2.5th and 97.5th percentiles. Therefore, an ideal PICP value for the learned model should be 95%. We employed 100 samples to approximate the estimated distribution, and all experiments were repeated 10 times, with mean and standard deviation recorded.

## C   MORE QUANTITATIVE RESULT

To assess the performance of TMDM in probabilistic multivariate time series forecasting, we applied our model to six datasets alongside several competitive baselines. Four metrics (QICE, PICP, CRPS,

---

[†]https://archive.ics.uci.edu/ml/datasets/ElectricityLoadDiagrams20112014

[‡]https://gis.cdc.gov/grasp/fluview/fluportaldashboard.html

[§]http://pems.dot.ca.gov/

[¶]https://www.bgc-jena.mpg.de/wetter/

and CRPS$_{sum}$) were employed as performance indicators for probabilistic forecasting. Additionally, we used two metrics (MSE and MAE) to evaluate other aspects of model performance. As shown in Table 6 and Table 7 , the NSformer integrated with our framework consistently achieved state-of-the-art (SOTA) performance across all benchmark datasets. Notably, when compared to previous SOTA results, TMDM achieved a remarkable 3.42 (from 71.12 to 74.54) improvement in PCIP for the Exchange dataset, 8.26 (from 79.57 to 87.83) improvement for ILI, 0.62 (from 72.58 to 73.2) improvement for ETTm2, 3.01 (from 79.34 to 82.35) improvement, b[i] for Electricity, 2.53 (from 84.3 to 86.83) improvement for Traffic, and 6.05 (from 66.92 to 72.97) improvement for Weather. It is worth noting that models like D$^3$VAE (Li et al., 2022) and TimeDiff (Shen & Kwok, 2023) were originally designed for point-to-point forecasting tasks, where probabilistic forecasting was not their primary focus. Consequently, their performance on QICE and PICP was suboptimal. However, they still demonstrated competitive performance in CRPS, CRPS$_{sum}$ (see Appendix C), MSE, and MAE (refer to Table 3). This underscores the importance of introducing new metrics (QICE and PICP) into the evaluation of probabilistic multivariate time series forecasting models.

Table 6: Performance comparisons on six real-world datasets in terms of CRPS-sum. The best results are boldfaced.

| Method | Exchange | ILI | ETTm2 | Electricity | Traffic | Weather |
|---|---|---|---|---|---|---|
| VAE | 7.19±0.178 | 9.84±0.289 | 7.91±0.150 | 9.56±0.235 | 7.65±0.119 | 8.98±0.142 |
| GP-Copula | 4.78±0.092 | 7.87±0.148 | 4.53±0.227 | 8.36±0.128 | 6.29±0.172 | 6.98±0.089 |
| Transformer-MAF | 4.21±0.162 | 6.28±0.248 | 3.57±0.390 | 6.27±0.177 | 5.23±0.128 | 6.19±0.281 |
| TimeGrad | 3.92±0.238 | 6.06±0.212 | 3.13±0.227 | 5.46±0.111 | 4.67±0.078 | 6.01±0.193 |
| CSDI | 4.20±0.153 | 6.98±0.065 | 3.07±0.130 | 4.54±0.368 | 3.98±0.136 | 5.39±0.183 |
| SSSD | 3.31±0.174 | 6.30±0.127 | 2.28±0.061 | 4.50±0.297 | 3.10±0.308 | 4.94±0.238 |
| D$^3$VAE | 2.05±0.197 | 6.50±0.138 | 1.93±0.066 | 3.80±0.290 | 3.23±0.136 | 3.82±0.196 |
| TimeDiff | 2.46±0.342 | 7.01±0.126 | 1.92±0.019 | 3.08±0.151 | 2.47±0.204 | 3.16±0.271 |
| ours | **1.87**±0.072 | **6.02**±0.109 | **1.75**±0.044 | **2.87**±0.118 | **2.10**±0.185 | **1.79**±0.086 |

Table 7: Performance comparisons on six real-world datasets in terms of PICP. The best results are boldfaced.

| Method | Exchange | ILI | ETTm2 | Electricity | Traffic | Weather |
|---|---|---|---|---|---|---|
| VAE | 64.62±3.060 | 68.31±2.365 | 67.18±1.648 | 70.56±2.362 | 78.50±5.538 | 58.92±3.215 |
| GP-Copula | 66.86±8.737 | 70.40±3.263 | 70.94±5.104 | 70.89±4.881 | 80.33±3.780 | 60.72±6.892 |
| Transformer-MAF | 69.74±10.706 | 73.68±6.853 | 71.23±1.106 | 73.89±6.111 | 80.96±1.585 | 61.30±7.185 |
| TimeGrad | 69.16±10.254 | 74.29±6.325 | 71.62±5.999 | 75.93±2.182 | 82.28±1.102 | 62.79±7.962 |
| CSDI | 69.21±9.997 | 76.18±4.932 | 71.78±2.686 | 78.94±3.574 | 83.51±6.593 | 62.71±4.832 |
| SSSD | 71.12±7.330 | 79.57±3.366 | 72.58±3.736 | 79.34±5.166 | 84.30±4.721 | 66.92±5.828 |
| D$^3$VAE | 21.38±5.034 | 5.20±0.723 | 13.20±1.401 | 33.96±1.411 | 7.12±2.882 | 20.44±5.575 |
| TimeDiff | 20.80±7.528 | 3.69±0.620 | 13.16±1.246 | 32.37±1.105 | 9.11±1.170 | 21.60±6.345 |
| ours | **74.54**±2.046 | **87.83**±7.215 | **73.20**±2.776 | **82.35**±3.538 | **86.83**±4.027 | **72.97**±2.583 |

Evaluating model performance with time-series data at different granularities is of significant importance in real-world applications. Our proposed TMDM can excel in such scenarios for the following reasons:

1. Similar to most time series forecasting models Zhou et al. (2021); Wu et al. (2021); Liu et al. (2022), we incorporate the actual timestamps as learnable time embeddings for each data point. Leveraging a well-designed Transformer, we can effectively capture the temporal correlations within the data. This design ensures that TMDM can adapt to various granularities of time-series data.

2. Within the diffusion model component, we also account for the time embedding in the data. This allows the model to generate multivariate time series with information from these embeddings, accommodating time series data at different granularities.

3. As shown in Table 1 in our paper, the selected datasets covered different granularities ranging from 10 minutes to 1 day, and TMDM demonstrated competitive performance across all databases.

This confirms the model's ability to handle situations where time-series data is available at different granularities.

4. To further evaluate TMDM's ability to handle varying time-series granularities, we conducted an experiment where we randomly removed $D$ data points from a given time series $y_{0:M}$ and used this modified dataset to test TMDM with the same settings as described in the paper. This challenging experiment simulates scenarios where the time intervals in multivariate time series vary, making it a rigorous test of the model's performance under changing granularities.

Table 8: Performance of TMDM on the time-series in different granularities.

| Dataset | Exchange | | ETTm2 | | Electricity | | Traffic | | Wether | |
| Metric | QICE | CRPS | QICE | CRPS | QICE | CRPS | QICE | CRPS | QICE | CRPS |
|---|---|---|---|---|---|---|---|---|---|---|
| TMDM-60 | 4.78 | 0.32 | 3.72 | 0.39 | 3.73 | 0.31 | 2.36 | 0.28 | 3.91 | 0.37 |
| TMDM-30 | 4.14 | 0.30 | 3.93 | 0.38 | 3.85 | 0.35 | 2.34 | 0.25 | 3.79 | 0.34 |
| TMDM | 4.38 | 0.32 | 3.75 | 0.37 | 3.81 | 0.33 | 2.36 | 0.26 | 3.87 | 0.36 |

As depicted in the table above (Table 8), TMDM-60 denotes our model with a prediction length of $192 + 60$, where we randomly remove 60 samples from the data. In this setting, TMDM is challenged to forecast multivariate time series (MTS) with varying granularities based on the time embedding. The results obtained from the 30 and 60 settings exhibit similar scores compared to the original setting, demonstrating the effectiveness of the proposed TMDM in accommodating time-series data with differing granularities.

The variation in the $CRPS_{sum}$ results of the baseline models compared to the published results is primarily due to differences in the experimental settings, specifically concerning the history length and prediction length. We have included the experiments on the settings in TimeGrand and CSDI in our paper, and the results can also be found as follows (Table 9 and Table 10 ):

Table 9: TimeGrand setting result.

| Dataset | Exchange | | Electricity | | Traffic | |
| Metric | QICE | $CRPS_{sum}$ | QICE | $CRPS_{sum}$ | QICE | $CRPS_{sum}$ |
|---|---|---|---|---|---|---|
| TimeGrand | 3.63 | 0.006 | 2.51 | 0.0206 | 1.91 | 0.044 |
| TMDM | 2.48 | 0.004 | 1.31 | 0.016 | 1.07 | 0.013 |

Table 10: CSDI setting result.

| Dataset | Exchange | | Electricity | | Traffic | |
| Metric | QICE | $CRPS_{sum}$ | QICE | $CRPS_{sum}$ | QICE | $CRPS_{sum}$ |
|---|---|---|---|---|---|---|
| TimeGrand | - | - | 2.59 | 0.021 | 2.02 | 0.044 |
| CSDI | - | - | 2.28 | 0.017 | 1.60 | 0.020 |
| TMDM | - | - | 1.36 | 0.014 | 1.23 | 0.015 |

CSDI occasionally matches or slightly surpasses TimeGrand, yet TMDM consistently outperforms both models across various datasets and metrics. TMDM showcases superior performance in probabilistic forecasting, reflected in its lower QICE and $CRPS_{SUM}$ values. These results emphasize TMDM's significant advancements in predictive accuracy and distributional modeling compared to TimeGrand and CSDI across diverse datasets and evaluation metrics.

# D   DERIVATION FOR FORWARD PROCESS POSTERIORS:

In this section, we derive the mean and variance of the forward process posteriors $q\left(\boldsymbol{y}_{0:M}^{t-1} \mid \boldsymbol{y}_{0:M}^{t-1}, \boldsymbol{y}_{0:M}^{0}, \hat{\boldsymbol{y}}_{0:M}\right)$ in Eq. 14:

$$q\left(\boldsymbol{y}_{0:M}^{t-1} \mid \boldsymbol{y}_{0:M}^{t-1}, \boldsymbol{y}_{0:M}^{0}, \hat{\boldsymbol{y}}_{0:M}\right) \propto q\left(\boldsymbol{y}_{0:M}^{t} \mid \boldsymbol{y}_{0:M}^{t-1}, \hat{\boldsymbol{y}}_{0:M}\right) q\left(\boldsymbol{y}_{0:M}^{t-1} \mid \boldsymbol{y}_{0:M}^{0}, \hat{\boldsymbol{y}}_{0:M}\right)$$

$$\propto \exp(-\frac{1}{2}(\frac{(\boldsymbol{y}_{0:M}^{t} - (1 - \sqrt{\bar{\alpha}^t})\hat{\boldsymbol{y}}_{0:M} - \sqrt{\bar{\alpha}^t}\boldsymbol{y}_{0:M}^{t-1})^2}{\beta^t}$$

$$+ \frac{(\boldsymbol{y}_{0:M}^{t-1} - \sqrt{\alpha^{t-1}}\boldsymbol{y}_{0:M}^{0} - (1 - \sqrt{\alpha^{t-1}})\hat{\boldsymbol{y}}_{0:M})^2}{1 - \alpha^{t-1}}))$$

$$\propto \exp(-\frac{1}{2}(\frac{\bar{\alpha}^t(\boldsymbol{y}_{0:M}^{t-1})^2 - 2\sqrt{\bar{\alpha}^t}(\boldsymbol{y}_{0:M}^{t} - (1 - \sqrt{\bar{\alpha}^t})\hat{\boldsymbol{y}}_{0:M})\boldsymbol{y}_{0:M}^{t-1}}{\beta^t}$$

$$+ \frac{(\boldsymbol{y}_{0:M}^{t-1})^2 - 2(\sqrt{\alpha^{t-1}}\boldsymbol{y}_{0:M}^{0} + (1 - \sqrt{\alpha^{t-1}})\hat{\boldsymbol{y}}_{0:M})\boldsymbol{y}_{0:M}^{t-1}}{1 - \alpha^{t-1}}))$$

$$= \exp(-\frac{1}{2}(A_1(\boldsymbol{y}_{0:M}^{t-1})^2 - 2A_2\boldsymbol{y}_{0:M}^{t-1}))$$

where

$$A_1 = \frac{\bar{\alpha}^t(1 - \alpha^{t-1}) + \beta^t}{\beta^t(1 - \alpha^{t-1})} = \frac{1 - \alpha^t}{\beta^t(1 - \alpha^{t-1})}$$

$$A_2 = \frac{\sqrt{\alpha^{t-1}}}{1 - \alpha^{t-1}}\boldsymbol{y}_{0:M}^{0} + \frac{\sqrt{\bar{\alpha}^t}}{\beta^t}\boldsymbol{y}_{0:M}^{t} + (\frac{\sqrt{\bar{\alpha}^t}(\sqrt{\bar{\alpha}^t} - 1)}{\beta^t} + \frac{1 - \sqrt{\alpha^{t-1}}}{1 - \alpha^{t-1}})\hat{\boldsymbol{y}}_{0:M}$$

and we have the posterior variance:

$$\tilde{\beta}^t = 1/A_1 = \frac{(1 - \alpha^{t-1})}{1 - \alpha^t}\beta^t$$

Meanwhile, the following coefficients of the terms in the posterior mean through dividing each coefficient in $A_2$ by $A_1$

$$\gamma_0 = \frac{\sqrt{\alpha^{t-1}}}{1 - \alpha^{t-1}}/A_1 = \frac{\beta^t\sqrt{\alpha^{t-1}}}{1 - \alpha^t}$$

$$\gamma_1 = \frac{\sqrt{\bar{\alpha}^t}}{\beta^t}/A_1 = \frac{(1 - \alpha^{t-1})\sqrt{\bar{\alpha}^t}}{1 - \alpha^t}$$

$$\gamma_2 = (\frac{\sqrt{\bar{\alpha}^t}(\sqrt{\bar{\alpha}^t} - 1)}{\beta^t} + \frac{1 - \sqrt{\alpha^{t-1}}}{1 - \alpha^{t-1}})/A_1$$

$$= \frac{\bar{\alpha}^t - \alpha^t - \sqrt{\bar{\alpha}^t}(1 - \alpha^{t-1}) + \beta^t - \beta^t\sqrt{\alpha^{t-1}}}{1 - \alpha^{t-1}}$$

$$= 1 + \frac{(\sqrt{\bar{\alpha}^t} - 1)(\sqrt{\bar{\alpha}^t} + \sqrt{\alpha^{t-1}})}{1 - \alpha^t}$$

which together give us the posterior mean

$$\boldsymbol{\mu}(\boldsymbol{y}_{0:M}^{0}, \boldsymbol{y}_{0:M}^{t}, \hat{\boldsymbol{y}}_{0:M}) = \gamma_0\boldsymbol{y}_{0:M}^{0} + \gamma_1\boldsymbol{y}_{0:M}^{t} + \gamma_2\hat{\boldsymbol{y}}_{0:M}$$

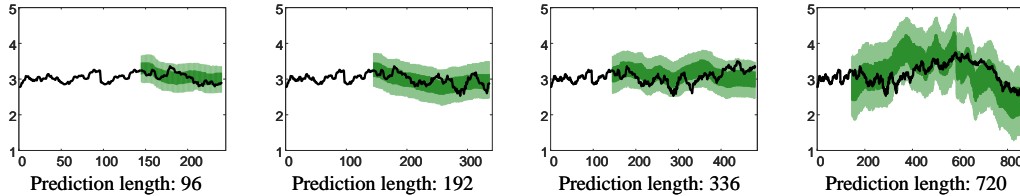

Figure 3: The prediction intervals in different predict lengths of the proposed TMDM

## E    DIFFERENT PREDICT LENGTHS COMPARISON OF PREDICTION INTERVALS

As shown in Fig. 3, we have presented prediction intervals generated by TMDM for various prediction lengths. In this experiment, we maintained a consistent history length, and it becomes evident that the prediction intervals widen as the prediction length extends. This indicates TMDM's ability to provide different levels of uncertainty when dealing with more challenge prediction tasks, a valuable characteristic for real-world applications.

## F    MORE COMPARISON OF PREDICTION INTERVALS FOR THE EXCHANGE AND WEATHER DATASET

Fig. 4, Fig. 5, and Fig. 6 offer further comparisons of prediction intervals for the Exchange and Weather Datasets. These figures include examples of both successful and challenging prediction cases.

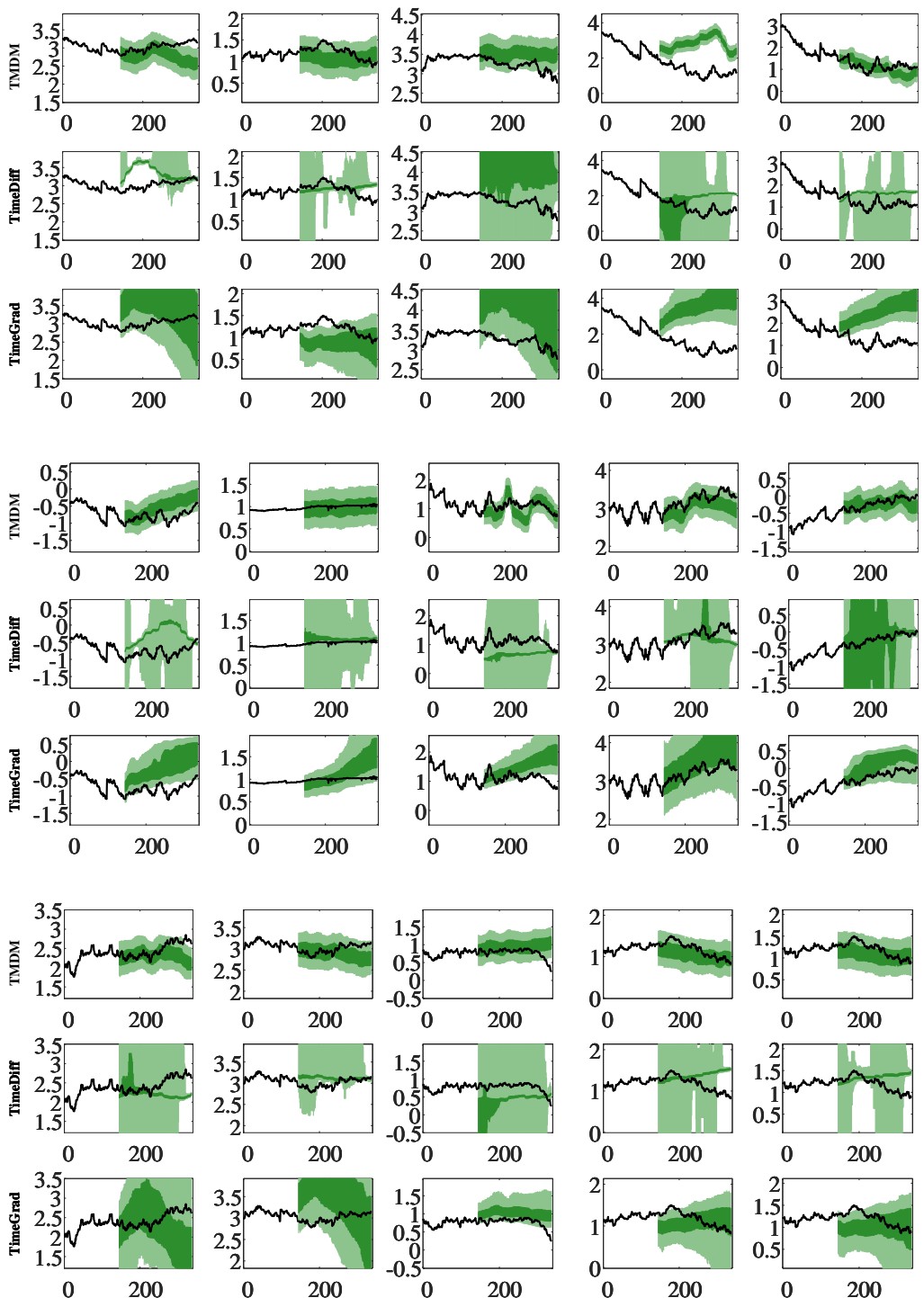

Figure 4: More comparison of prediction intervals for the Exchange Dataset. We display the predicted median and visualize the 50% and 90% distribution intervals, the black line representing the test set ground-truth.

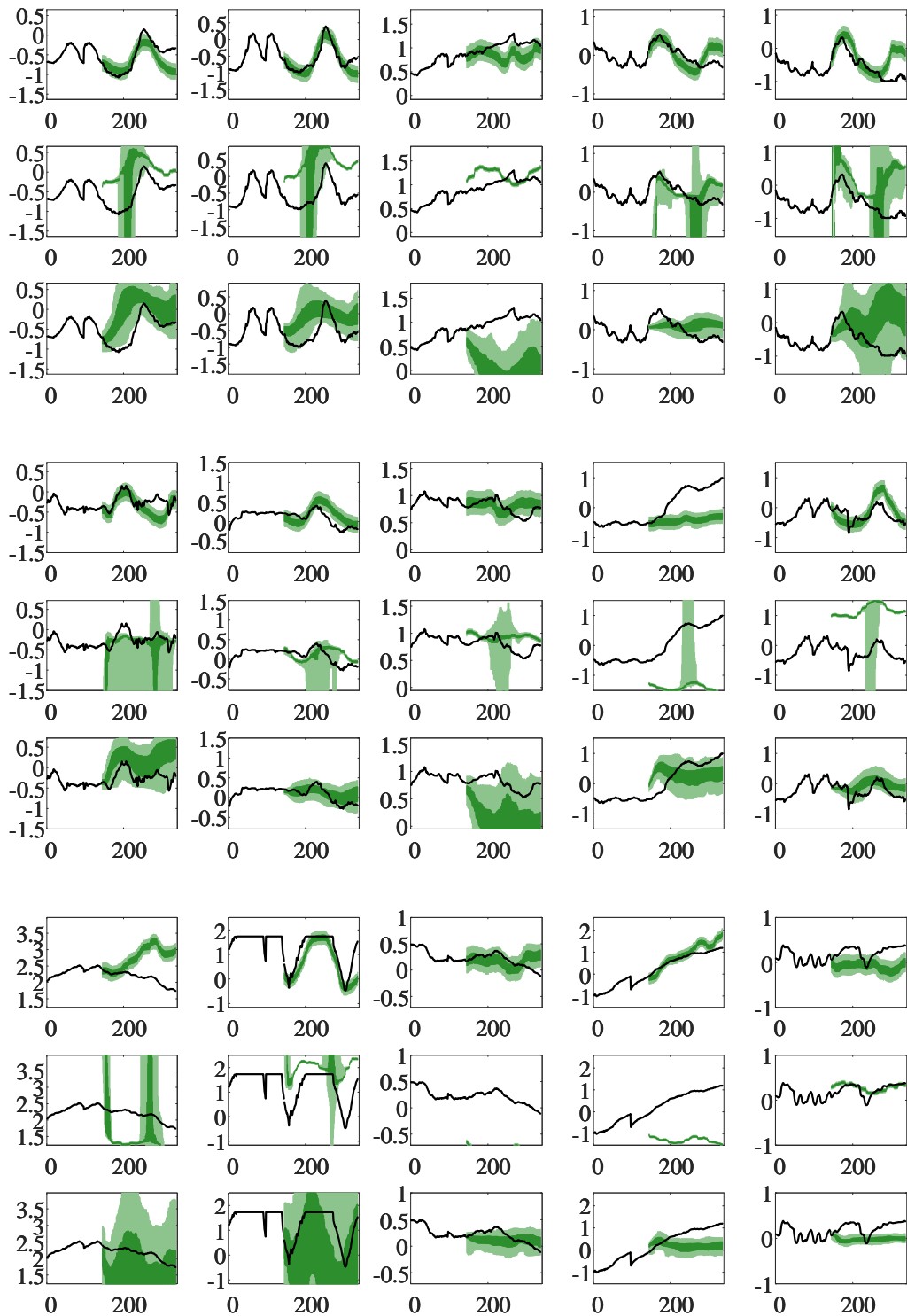

Figure 5: More comparison of prediction intervals for the Weather Dataset. We display the predicted median and visualize the 50% and 90% distribution intervals, the black line representing the test set ground-truth.

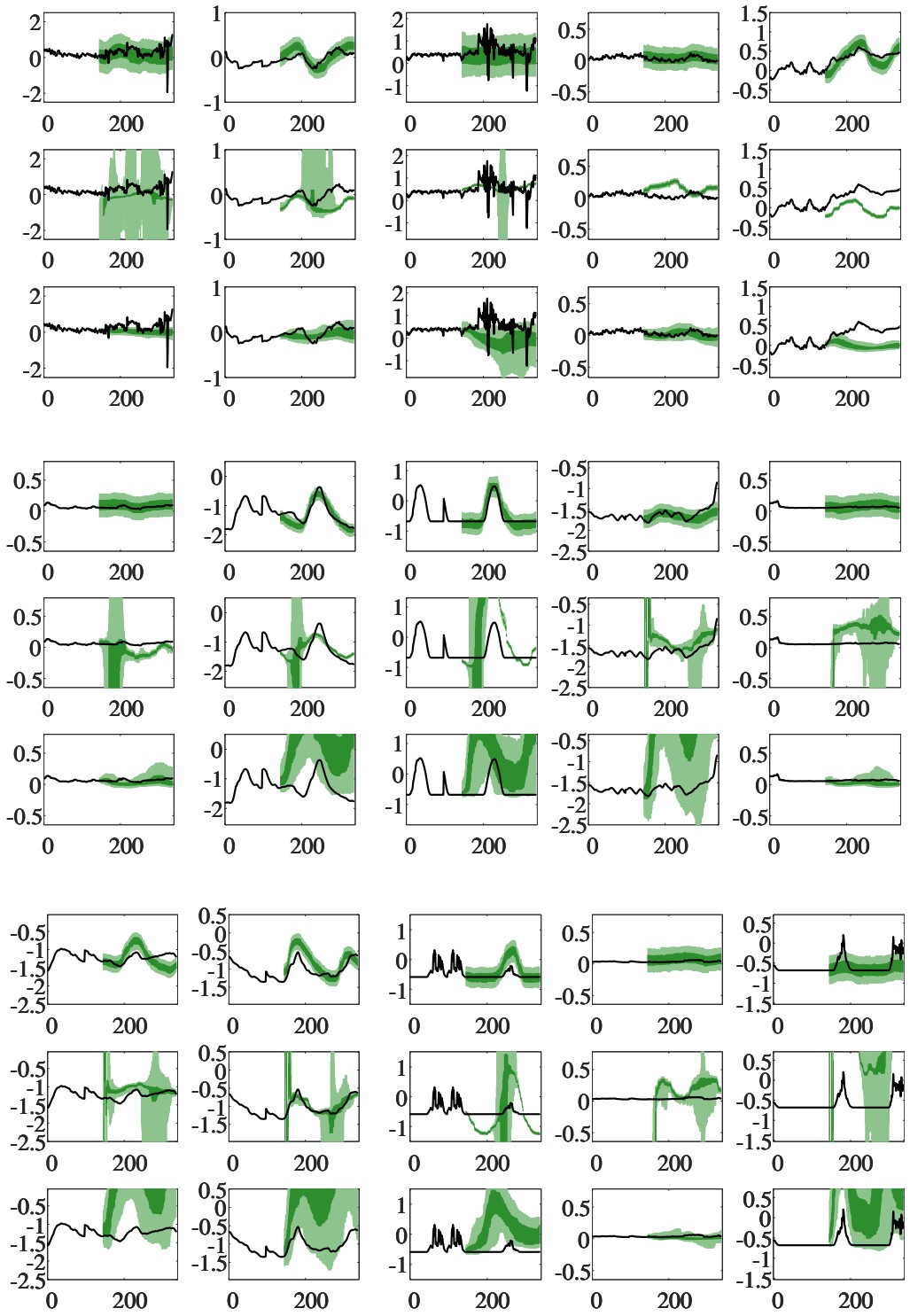

Figure 6: More comparison of prediction intervals for the Weather Dataset. We display the predicted median and visualize the 50% and 90% distribution intervals, the black line representing the test set ground-truth.

