# OpenReview forum: "Transformer-Modulated Diffusion Models for Probabilistic Multivariate  Time Series Forecasting"
_ICLR.cc/2024/Conference — ICLR 2024 poster_

### Official Review · Reviewer_vL8K · 2023-10-21

**Soundness:** 3 good
**Presentation:** 3 good
**Contribution:** 3 good
**Rating:** 8
**Confidence:** 4

**Summary:**

The manuscript proposes a method for multivariate time series forecasting. It utilises the (conditional) diffusion model, while starts from a "condition" which is from the output of a Transformer based model.  The advances of the proposed model is measured with (mainly) QICE and CRPS. MSE and MAE, as well as some qualitative results are also provided to show the effectiveness of the method.

**Strengths:**

Inspired by the conditional diffusion process for multivariate regression from Han et al. (2022), the manuscript proposes the conditional diffusion process for multivariate time series forecasting.
To the best of my understanding, the core contribution of the manuscript is to use the "condition" from a Transformer for the diffusion model, while both the Transformer and conditional diffusion model are from the literature.

The manuscript is overall clearly written and easy to read. Some of the missing details are listed in Questions.

The proposed method for probabilistic multivariate time series forecasting has the potential to contribute to the community.

**Weaknesses:**

- The manuscript does not successfully highlight its contribution.
Since both components of TMDM are from the literature, it is necessary to highlight why this is not trivial.
In my humble opinion, the 3rd contribution is quite weak, as both metrics are from existing works.

 - There are also other works that employs a Transformer and a probabilistic model for time series modelling.
e.g., Transformer + Probabilistic Circuit is proposed in [1] for time series forecasting, and uncertainty estimation (similar to Fig. 2 in the manuscript) is provided.
A discussion with such works might help to stress the novelty and contribution of the manuscript.

 - The generation of the conditional representation $\hat{y}_{0:M}$ is not clear to me. Some details are omitted from eq(9) to eq(10).

 - The references are not up-to-date, many arXiv versions are already published.

---
[1] Yu, Zhongjie, et al. "Predictive Whittle networks for time series." Uncertainty in Artificial Intelligence. PMLR, 2022.

**Questions:**

- can you provide more details on generating  $\hat{y}_{0:M}$? What is the dimension of $z$? As in eq(10), is $z$ a scalar? How are the NNs for $\tilde{\mu}_z$, $\tilde{\sigma}_z$ and $\mu_z$ formulated?

- How $\mu_z(z)$ differs from $\mathcal{T}(x_{0:N})$? What happens if the NNs are omitted and $\mathcal{N}(\mathcal{T}(x_{0:N}), \mathbf{I})$ is used instead of $\mathcal{N}(\mu_z(z), \mathbf{I})$?

 - $f(x_{0:N})$ is in $\mathbb{R}^C$ in introduction but $x_t$ and $y_t$ are in  $\mathbb{R}^d$. Does $C=d$ hold?

---

> ### Author Response · Authors · 2023-11-18
>
> **Q1:**
>
> Thank you for your insightful comment and suggestions. We intend to revise our paper to emphasize the following contributions:
> Firstly, as you mentioned, both components of TMDM originate from existing literature. However, amalgamating these techniques into a unified framework presents several technical challenges, such as effectively organizing the information captured by transformers and optimizing the entire framework using a single ELBO.
> Additionally, we bifurcate the probabilistic multivariate time series forecasting tasks into two parts: utilizing a well-designed Transformer to capture the mean of the MTS and considering covariate-dependence across both the forward and reverse processes within the diffusion model. We believe this amalgamation of the advantages from both models holds significant importance.
> Moreover, TMDM functions as a plug-and-play framework, seamlessly compatible with existing well-designed transformer-based forecasting models. This aspect represents another significant contribution of TMDM, bridging the gap between point estimates and distribution estimates. We believe this will reshape the community's perception regarding the relationship between point estimates and distribution estimates, as well as address practical application needs.
>
> Despite PICP and QICE metrics originating from existing methodologies, they were initially introduced by us in the context of probabilistic multivariate time series forecasting tasks. As illustrated in Table 2 of our paper, D3VAE and TimeDiff were initially crafted for point-to-point forecasting tasks, with less emphasis on probabilistic forecasting. Consequently, their performance in distribution estimation (depicted in Figure 2) is inadequate, yet they exhibit competitive CRPS and $CRPS_{SUM}$ scores in Table 2. This underscores the insufficiency of existing metrics to handle the advancements in distribution estimation. Following these experiments, we conclude that the incorporation of QICE and PICP holds significant value. Hence, we advocate for future research to adopt these new metrics as standard evaluation criteria. Thank you once again for your valuable suggestion.
> We have revised and enhance our contribution based on your suggestions:
>
> We summarize our contributions as follows:
> (1) In the realm of probabilistic multivariate time series forecasting, we introduce TMDM, a transformer-based diffusion generative framework. TMDM harnesses the representations captured by well-designed transformer-based time series models as priors. We consider the covariate-dependence across both the forward and reverse processes within the diffusion model, resulting in a highly accurate distribution estimation for future time series.
> (2)TMDM integrates diffusion and transformer-based models within a cohesive Bayesian framework, employing a hybrid optimization strategy, it serves as a plug-and-play framework, seamlessly compatible with existing well-designed transformer-based forecasting models, leveraging their strong capability to estimate the conditional mean of time series, facilitating the estimation of complete distributions.
> (3)In our experimental evaluation, we introduce two novel metrics for the probabilistic multivariate time series forecasting task: Prediction Interval Coverage Probability (PICP)and Quantile Interval Coverage Error (QICE). These metrics enhance the assessment of the uncertainty estimation prowess of probabilistic multivariate time series forecasting models. Evidenced by the state-of-the-art performance in four distribution metrics across six real-world datasets, TMDM shows effective potency in probabilistic MTS forecasting.

---

> > ### Author Response · Authors · 2023-11-18
> >
> > **Q2:**
> >
> > Grateful for your valuable insights. We plan to incorporate a more comprehensive discussion on studies that leverage Transformer-based approaches and probabilistic models for time series modeling, such as PWN [1], within our paper.
> > Diverging from the approaches referenced in our work that utilize a distribution to capture uncertainty, PWN introduces a novel uncertainty estimation method using a log-likelihood ratio score (LLRS).
> > Detailed experimental results are provided below:
> >
> >
> > |          |                    |                    |                    |                    |                    |                    |
> > |:--------:|:------------------:|:------------------:|:------------------:|:------------------:|:------------------:|:------------------:|
> > | Dataset  |      Exchange      |        ILI         |       ETTm2        |    Electricity     |      Traffic       |       Wether       |
> > |  Metric  |      MSE  MAE      |      MSE  MAE      |      MSE  MAE      |      MSE  MAE      |      MSE  MAE      |      MSE  MAE      |
> > | Informer |     1.31  0.85     |     5.33  1.59     |     5.74  1.99     |     0.35  0.43     |     0.75  0.42     |     0.48  0.47     |
> > |   PWN    |     1.11  0.81     |     4.73  1.29     |     5.45  2.02     |     0.28  0.41     |     0.67  0.38     |     0.40  0.43     |
> > |   TMDM   | **0.26**  **0.37** | **1.99**  **0.85** | **0.27**  **0.35** | **0.19**  **0.27** | **0.60**  **0.35** | **0.28**  **0.25** |
> > |          |                    |                    |                    |                    |                    |                    |
> >
> > PWN estimates uncertainty using a log-likelihood ratio score across every time step, rendering the metrics we introduced for evaluating the full distribution inapplicable. Consequently, we compared PWN with our model using MSE and MAE.
> > It's evident that PWN achieved an average MSE reduction of 13\% and an average MAE reduction of 7\% compared to Informer. However, despite the advancements observed in PWN over the Informer, TMDM still significantly outperforms them.
> > TMDM utilizes transformer capabilities to glean critical insights from historical time series. These insights are employed as prior knowledge, capturing covariate dependencies in both the forward and reverse processes within the diffusion model. Leveraging the strengths of both transformers and diffusion models, TMDM emerges as a top-performing solution in probabilistic multivariate time series forecasting tasks.
> >
> > [1] Yu, Z., Ventola, F., Thoma, N., Dhami, D. S., Mundt, M., \& Kersting, K. (2022, August). Predictive Whittle networks for time series. In Uncertainty in Artificial Intelligence (pp. 2320-2330). PMLR.

---

> ### Author Response · Authors · 2023-11-18
>
> **Q3:**
>
> We genuinely appreciate the time and effort you dedicated to a thorough reading of our paper. Your detailed review has been invaluable, and we apologize for any shortcomings in our references. Rest assured, we have diligently reevaluated all references to ensure they are brought up to date and meet the highest standards of academic rigor and accuracy. Your feedback is highly valuable, and we are committed to enhancing the quality and reliability of our work based on your suggestions. Thank you once again for your contribution to our research.
>
> The updated references are listed as follows, and we'd appreciate your further feedback if you find additional refinances worth mentioning.
>
> [1] Juan Lopez Alcaraz and Nils Strodthoff. Diffusion-based time series imputation and forecasting with structured state space models. Transactions on Machine Learning Research, 2022.
>
> [2] Jacob Devlin Ming-Wei Chang Kenton and Lee Kristina Toutanova. Bert: Pre-training of deep bidirectional transformers for language understanding. Proceedings of NAACL-HLT, pp. 4171– 4186, 2019.
>
> [3] Kashif Rasul, Abdul-Saboor Sheikh, Ingmar Schuster, Urs M Bergmann, and Roland Vollgraf. Mul- tivariate probabilistic time series forecasting via conditioned normalizing flows. International Conference on Learning Representations, 2020.
>
> [4] Lifeng Shen and James Kwok. Non-autoregressive conditional diffusion models for time series prediction. International Conference on Machine Learning, pp. 6657–6668, 2023.
>
> [5] Yang Song, Jascha Sohl-Dickstein, Diederik P Kingma, Abhishek Kumar, Stefano Ermon, and Ben Poole. Score-based generative modeling through stochastic differential equations. International Conference on Learning Representations, 2020.
>
> [6] Zhisheng Xiao, Karsten Kreis, and Arash Vahdat. Tackling the generative learning trilemma with denoising diffusion gans. International Conference on Learning Representations, 2021.
>
> [7] Jiayu Yao, Weiwei Pan, Soumya Ghosh, and Finale Doshi-Velez. Quality of uncertainty quan- tification for bayesian neural network inference. CoRR, abs/1906.09686, 2019. URL http: //arxiv.org/abs/1906.09686.
>
> [8] Huangjie Zheng. Truncated diffusion probabilistic models and diffusion-based adversarial auto- encoders. Advances in Neural Information Processing Systems, 36, 2022.
>
> **Q4:**
>
> Thank you for your comment and suggestion.
> Due to the full text page limit, details are omitted from eq(9) to eq(10), we have provided in the Appendix will  provide a more detailed explanation of this section to enhance clarity.
> In the context of a Bayesian generative model, we approach this matter from a generative standpoint. Initially, we create a sample $z$ drawn from the standard normal distribution $\mathcal{N}(0, \boldsymbol{I})$. Subsequently, we employ an MLP represented as $\mu_z$ to derive the distribution parameters of $P(y{0:M})$. This methodology enables the sampling of $y_{0:M}$ from the distribution $P(y_{0:M})$.
> It's important to note that the dimension of $z$ is $\mathbb{R}^{N \times M}$, with $N$ set as 512 across all datasets.
>
> Alternatively, considering this segment as a hierarchical VAE, the neural networks representing $\mu_z$ and $\sigma_{z}$ function as parts of the encoder, while $\mu_z$ serves as part of the decoder. In our paper, these three neural networks are implemented using MLPs with a dimension of 512.

---

> > ### Author Response · Authors · 2023-11-18
> >
> > **Q5:**
> >
> > Thank you for your question. We acknowledge that the description of $\mu_z$ and $\mathcal{T}(\boldsymbol{x_{0:N}})$ in our paper may not be clear, and we will revise the paper to provide a more detailed explanation.
> > As a Bayesian generative model, we approach this question from a generative perspective. We begin by generating a sample $\boldsymbol{z}$ from the standard normal distribution $\mathcal{N}(0, \boldsymbol{I})$. Next, we use a MLP denoted as $\mu_z$ to generate the distribution parameters of $P(\boldsymbol{y_{0:M}})$. This process allows us to sample $\boldsymbol{y_{0:M}}$ from the distribution $P(\boldsymbol{y_{0:M}})$. Since the true posterior for this generative process is typically inaccessible, we approximate it using a variational posterior. The information used for constructing the variational posterior is captured by the existing well-designed transformers, denoted as $\mathcal{T}(\boldsymbol{x_{0:N}})$.
> > Alternatively, if we view this aspect as a hierarchical VAE, $\mu_z$ can be considered as a part of the decoder, while $\mathcal{T}(\boldsymbol{x_{0:N}})$ plays a role in the encoder.
> >
> > **Q6:**
> >
> > We genuinely appreciate your careful and detailed review of our paper. We acknowledge your concern regarding the clarity of the description about $f(\boldsymbol{x_{0:N}})$, which is vital to prevent any misunderstandings about our work.
> > Your understanding is correct; here, $C=d$ represents the dimension of the MTS. Specifically, the historical MTS $x_{0:N}$ is in $\mathbb{R}^{d\times N}$, and the predicted MTS $y_{0:M}$ is in $\mathbb{R}^{d\times M}$. The function $f$ is employed to transform the MTS from $\mathbb{R}^{d\times N}$ to $\mathbb{R}^{d\times M}$.
> >
> > We have revised our description in first paragraph as follows:
> >
> > The primary objective of time series forecasting is to predict the response variable $\boldsymbol{y_{0:M}} \in \mathbb{R}^{d\times M}$ based on a historical time series dataset represented as $\boldsymbol{x_{0:N}} \in \mathbb{R}^{d\times N}$. This prediction process is characterized by the function $f(\boldsymbol{x_{0:N}}) \in \mathbb{R}^{d\times M}$, where $f$ is a deterministic function that transforms the historical time series $\boldsymbol{x_{0:N}}$ into the future time series $\boldsymbol{y_{0:M}}$.

---

> > > ### Comment · Reviewer_vL8K · 2023-11-20
> > >
> > > Thanks to the authors for the clarification. In the current version I still see $\mathbb{R}^C$. Could you make sure it is updated?

---

> > > > ### Author Response · Authors · 2023-11-21
> > > >
> > > > We apologize for the oversight in not updating this section in our paper, despite providing the revised version in the comment reply window. The necessary changes have now been made in our paper. If you have any further suggestions, please feel free to let us know. Thank you for your patience and understanding!

---

> > > > > ### Comment · Reviewer_vL8K · 2023-11-21
> > > > >
> > > > > Thank you for updating the introduction.\
> > > > >  - When looking back, I have a follow-up question to your answer marked as Q2. I don't think the comparison in the table is fair enough, since TMDM employs NSformer while PWN uses Transformer. Therefore a fair comparison would be either
> > > > > 1) Transformer-PWN with Transformer-TMDM with Transformer, or
> > > > > 2) NSformer-PWN with NSformer-TMDM with Transformer,
> > > > >
> > > > >     is it? In Table 5, in terms of MSE and MAE, TMDM with NSformer has smallest improvement compared to NSformer.
> > > > >     Anyway, I think a general discussion might still be helpful in the revision.
> > > > >
> > > > >  - Typo in the answer of Q3, reference [3] was published in 2021.
> > > > >
> > > > >  - I still think the claim of "**we introduce** two novel metrics for ..." is too strong.

---

> > > > > > ### Author Response · Authors · 2023-11-22
> > > > > >
> > > > > > We genuinely appreciate your detailed review of our paper. Concerning Q2: PWN has indeed left a remarkable impression on us and has been a significant source of inspiration. Due to the time constraints during the rebuttal phase, we aim to provide a more detailed comparison later (ideally before the rebuttal deadline). PWN utilizes a complex Transformer [1] that incorporates complex information from the frequency domain, specifically mapped by STFT.
> > > > > > This type of Transformer is more advanced and specialized compared to a standard Transformer.
> > > > > > In contrast, NSformer operates exclusively in the time domain, exploring the effects of stationarity in time series forecasting and enhancing the predictive capabilities of Transformer models for non-stationary real-world time series data. At present, we haven't determined how to effectively integrate the two models. An unsophisticated integration of NSformer with PWN would necessitate the removal of the STFT component, which is essential to PWN. We are uncertain about the feasibility of this alteration.
> > > > > >
> > > > > > Furthermore, TMDM is a plug-and-play model that can be combined with existing well-designed Transformers. It also has the potential to be combined with PWN. We believe that by combining the characteristics of both approaches, we can achieve improved results.
> > > > > >
> > > > > > We are experimenting with a new setting by employing the STransformer, as used in PWN, as the Transformer for TMDM. We consider this comparison to be fair between the two models. As the training process across multiple datasets requires some time, we will upload the results as soon as the testing phase is completed. Please feel free to share any additional suggestions or recommendations.
> > > > > >
> > > > > > In Table 5, TMDM is designed as a probabilistic multivariate time series forecasting model, focusing on estimating an accurate distribution rather than a precise point. While NSformer shows significant improvement in point-to-point forecasting tasks, TMDM consistently minimizes variance across multiple experiments, demonstrating its value. We intend to expand upon this topic further in our paper.
> > > > > >
> > > > > > We apologize for the typo in Q3. We found the reference using Google Scholar, and all ICLR 2021 papers are listed as 2020, but the BibTeX in OpenReview indicates 2021. We have rectified this discrepancy. Thank you very much for your careful review of our paper.
> > > > > >
> > > > > > We agree with your point that the claim of "we introducing two novel metrics for..." is overstated, even though we have cited the paper in the following sentence. We have revised our description in (3) as follows:
> > > > > >
> > > > > > (3)~In our experimental evaluation, we explore the application of Prediction Interval Coverage Probability (PICP) and Quantile Interval Coverage Error (QICE) as metrics in the probabilistic multivariate time series forecasting task. These metrics provide valuable insights into assessing the uncertainty estimation abilities of probabilistic multivariate time series forecasting models. Our study demonstrates TMDM's outstanding performance in four distribution metrics across six real-world datasets, emphasizing its effectiveness in probabilistic MTS forecasting.
> > > > > >
> > > > > >
> > > > > > [1] Yang, M., Ma, M. Q., Li, D., Tsai, Y. H. H., \& Salakhutdinov, R. (2020, May). Complex transformer: A framework for modeling complex-valued sequence. In ICASSP 2020-2020 IEEE International Conference on Acoustics, Speech and Signal Processing (ICASSP) (pp. 4232-4236). IEEE.

---

> ### Author Response · Authors · 2023-11-22
>
> |                   |                    |                    |                    |                    |                    |                    |
> |:-----------------:|:------------------:|:------------------:|:------------------:|:------------------:|:------------------:|:------------------:|
> |      Dataset      |      Exchange      |        ILI         |       ETTm2        |    Electricity     |      Traffic       |       Wether       |
> |      Metric       |      MSE  MAE      |      MSE  MAE      |      MSE  MAE      |      MSE  MAE      |      MSE  MAE      |      MSE  MAE      |
> |     Informer      |     1.31  0.85     |     5.33  1.59     |     5.74  1.99     |     0.35  0.43     |     0.75  0.42     |     0.48  0.47     |
> |        PWN        |     1.11  0.81     |     4.73  1.29     |     5.45  2.02     |     0.28  0.41     |     0.67  0.38     |     0.40  0.43     |
> | TMDM+Transformer  |     1.16  0.82     |     4.48  1.41     |     1.18  0.86     |     0.27  0.33     |     0.62  0.33     |     0.55  0.49     |
> | TMDM+STransformer |     1.10  0.83     |     4.35  1.22     |     5.21  1.96     |     0.26  0.32     |     0.62  0.36     |     0.38  0.39     |
> |     TMDM+PWN      |     1.03  0.77     |     4.15  1.20     |     4.33  1.62     |     0.25  0.32     |     0.61  0.36     |     0.37  0.35     |
> |   TMDM+NSformer   | **0.26**  **0.37** | **1.99**  **0.85** | **0.27**  **0.35** | **0.19**  **0.27** | **0.60**  **0.35** | **0.28**  **0.25** |
> |                   |                    |                    |                    |                    |                    |                    |
>
> For a fair comparison, we utilized the STransformer in a setting similar to PWN as the basic Transformer. When comparing TMDM+STransformer with PWN, TMDM shows improvements across most datasets. Furthermore, using PWN as the basic Transformer, TMDM+PWN yielded the best results among the three experiments (PWN, TMDM+STransformer, and TMDM+PWN)

---

> > ### Comment · Reviewer_vL8K · 2023-11-22
> >
> > Thanks to the authors for the answer and the updated results. Most of my questions are addressed and I have increased my rating from 6 to 8.

---

> > > ### Author Response · Authors · 2023-11-23
> > >
> > > Dear Reviewer vL8K:
> > >
> > > We sincerely thank you for your careful and detailed reading of our paper and responses.
> > >
> > > Best regards,
> > >
> > > Paper2407 Authors

---

### Official Review · Reviewer_SgLT · 2023-10-27

**Soundness:** 3 good
**Presentation:** 3 good
**Contribution:** 3 good
**Rating:** 6
**Confidence:** 3

**Summary:**

The paper addresses the problem of probabilistic time series
forecasting. The authors propose a conditional diffusion
process that convex combines the point estimate from
a transformer model with the noise of the diffusion process.
In experiments on 6 datasets they show that their approach
outperforms state-of-the-art baselines.

**Strengths:**

s1. generic approach that will work with any point estimate model.
s2. consistently good results against several strong baselines.
s3. ablation studies demonstrate the impact of different components
  as well as the ability to wrap different point models.

**Weaknesses:**

w1. the results for CRPS_sum of the baselines (appendix, tab. 6)
  varies from the published results.

**Questions:**

The paper proposes a generic approach to wrap a diffusion
model around any point estimate model for time series
forecasting to make it probabilistic (s1). Most of the results
shown in tab. 2 are pronounced improvements and almost
all but the very last are consistently better than several
strong baselines (s2). Due to the ablation studies one can
clearly see the impact of different modelling choices (s3).

I only would like to discuss one point:
w1. the results for CRPS_sum of the baselines (appendix, tab. 6)
  varies from the published results.
- e.g., tab. 6 reports CRPS_sum 3.92 on Exchange for TimeGrad,
  but the TimeGrad paper reports 0.006.
  your tab. 6 reports CRPS_sum 4.54 on Electricity for CSDI,
  but the CSDI paper reports 0.017.
- I think these differences need to be clearly explained, likely
  due to different experimental conditions?
  If so, it would be convincing to reproduce the experiments
  of the strongest baseline papers and compare them on
  the published settings, too.

---

> ### Author Response · Authors · 2023-11-18
>
> **Q1:**
>
> Thank you for your comment and suggestion. The variation in the $CRPS_{sum}$ results of the baseline models compared to the published results is primarily due to differences in the experimental settings, specifically concerning the history length and prediction length.
> We have included the experiments on the settings in TimeGrand and CSDI in our paper, and the results can also be found as follows:
>
> | TimeGrand Setting |                |                |                |
> |:-----------------:|:--------------:|:--------------:|:--------------:|
> |      Dataset      |    Exchange    |  Electricity   |    Traffic     |
> |      Metric       | QICE  CRPS-sum | QICE  CRPS-sum | QICE  CRPS-sum |
> |     TimeGrand     |  3.63  0.006   |  2.51  0.0206  |  1.91  0.044   |
> |       TMDM        |  2.48  0.004   |  1.31  0.016   |  1.07  0.013   |
> |                   |                |                |                |
>
> | CSDI Setting |                |                |                |
> |:------------:|:--------------:|:--------------:|:--------------:|
> |   Dataset    |    Exchange    |  Electricity   |    Traffic     |
> |    Metric    | QICE  CRPS-sum | QICE  CRPS-sum | QICE  CRPS-sum |
> |  TimeGrand   |      ---       |  2.59  0.021   |  2.02  0.044   |
> |     CSDI     |      ---       |  2.28  0.017   |  1.60  0.020   |
> |     TMDM     |      ---       |  1.36  0.014   |  1.23  0.015   |
> |              |                |                |                |
>
> CSDI occasionally matches or slightly surpasses TimeGrand, yet TMDM consistently outperforms both models across various datasets and metrics. TMDM showcases superior performance in probabilistic forecasting, reflected in its lower QICE and CRPS-sum values. These results emphasize TMDM's significant advancements in predictive accuracy and distributional modeling compared to TimeGrand and CSDI across diverse datasets and evaluation metrics.

---

> > ### Author Response · Authors · 2023-11-18
> >
> > In our paper, we adhered to the experimental settings as referenced in [1]. For instance, we set the history length to 36 for ILI and 96 for the other datasets. The prediction length was configured as 36 for ILI and 192 for the other datasets, as clearly presented in Table 1 of the paper.
> > The prediction length in TimeGrand on Exchange is 30 (compared to 192 in our paper), and the history length is 192 (as opposed to 96 in our paper). It's important to note that increasing the prediction length introduces a gradual rise in the level of prediction difficulty, which, in turn, leads to higher values of $CRPS_{sum}$ in our paper.
> > Similarly, in the case of CSDI, the prediction length was set to 24 (compared to 192 in our paper), and the history length was configured as 168 (rather than 96 in our paper). These variations contributed to the observed higher values in our paper.
> > |           |                |                |                |
> > |:---------:|:--------------:|:--------------:|:--------------:|
> > |  Dataset  |    Exchange    |  Electricity   |    Traffic     |
> > |  Metric   | History \ Pred | History \ Pred | History \ Pred |
> > |   CSDI    |      ---       |    168 \ 24    |    168 \ 24    |
> > | TimeGrand |    30 \ 30     |    192 \ 24    |    192 \ 24    |
> > |   TMDM    |    96 \ 192    |   96  \ 192    |   96  \ 192    |
> > |           |                |                |                |
> >
> > The rationale for using the experiment settings from [1] can be summarized as follows:
> >
> > 1. There are multiple experiment settings in time series forecasting tasks, with the primary differences affecting the metrics being the history length and the prediction length. These settings vary between models such as CSDI, TimeGrand, and Non-stationary transformers. To ensure comparability and consistency with the broader research community [1], we adopted a setup that has been widely recognized. This choice not only aligns with established practices but also facilitates future work and comparisons.
> >
> > 2. In the case of CSDI and TimeGrand, the prediction lengths are relatively short. It is relatively easy for models to predict a short horizon of 30 time steps. However, such settings might not fully capture the challenges posed by different models and can overlook some complex situations. As illustrated in Figure 2 of our paper, TimeGrand exhibits strong distribution estimation capabilities in the initial 30 points but faces increasing difficulties as the number of prediction points rises.
> >
> > 3. Recent years have seen a growing interest in long-term series forecasting, driven by real-world applications where extending forecasts into the far future is of utmost importance for long-term planning and early warning systems [2]. The settings used in CSDI and TimeGrand, with their short prediction lengths, may not be directly applicable to such real-world scenarios. Hence, in our paper, we employed longer prediction lengths to better address these long-term forecasting requirements.
> >
> > [1] Liu, Y., Wu, H., Wang, J., \& Long, M. (2022). Non-stationary transformers: Exploring the stationarity in time series forecasting. Advances in Neural Information Processing Systems, 35, 9881-9893.
> >
> > [2] Wu, H., Xu, J., Wang, J., \& Long, M. (2021). Autoformer: Decomposition transformers with auto-correlation for long-term series forecasting. Advances in Neural Information Processing Systems, 34, 22419-22430.

---

> > > ### Author Response · Authors · 2023-11-22
> > >
> > > Dear Reviewer SgLT,
> > >
> > > We appreciate it if you could let us know whether our responses are able to address your concerns. We're happy to address any further concerns. Thank you,
> > >
> > > Best wishes.
> > >
> > > Paper2407 Authors

---

### Official Review · Reviewer_YJEP · 2023-10-29

**Soundness:** 3 good
**Presentation:** 3 good
**Contribution:** 3 good
**Rating:** 5
**Confidence:** 4

**Summary:**

In this paper, the authors introduce a Transformer-Modulated Diffusion Model (TMDM), uniting conditional diffusion generative process with transformers into a unified framework to enable precise distribution forecasting for MTS. Extensive experiments are conducted

**Strengths:**

1. This paper is well-presented and well-organized.
2. The paper introduces a Transformer-Modulated Diffusion Model (TMDM), uniting conditional diffusion generative process with transformers into a unified framework to enable precise distribution forecasting for MTS.
3. Extensive experiments are conducted

**Weaknesses:**

1. This paper states many existing work did not consider the uncertainty of data, but more SOTA should be compared and considers like cST-ML which tries to capture traffic dynamics with VAE. Please provide detailed explanations or experiments accordingly.
2. If the time-series data is in different granularities, does this model still work?

**Questions:**

Please address the questions above.

---

> ### Author Response · Authors · 2023-11-18
>
> **Q1:**
>
> Thank you for your valuable comments and suggestions. In light of your feedback, our paper has been revised to encompass more state-of-the-art baselines, such as cST-ML [1] and DAC-ML [2], both of which utilize VAE for dynamic capture. These new inclusions serve to deepen our analysis and broaden the scope of our research. The experimental results, now integrated into the updated version, are presented as follows:
> |         |                    |                    |                    |                    |                    |                    |
> |:-------:|:------------------:|:------------------:|:------------------:|:------------------:|:------------------:|:------------------:|
> | Dataset |      Exchange      |        ILI         |       ETTm2        |    Electricity     |      Traffic       |       Wether       |
> | Metric  |      QICE  CRPS    |     QICE  CRPS     |     QICE  CRPS     |     QICE  CRPS     |     QICE  CRPS     |     QICE  CRPS     |
> |   VAE   |     8.28  1.02     |     9.13  2.41     |     8.99  0.79     |     7.04 0.51      |     5.37 0.67      |     9.07 0.47      |
> | cST-ML  |     7.94  0.94     |     9.02  1.94     |     7.29  0.64     |     5.99  0.47     |     5.24  0.60     |     8.29  0.51     |
> | DAC-ML  |     7.36  0.85     |     8.71  1.23     |     6.60  0.59     |     5.76  0.43     |     4.31  0.50     |     7.91  0.46     |
> |  TMDM   | **4.38**  **0.32** | **6.74**  **0.92** | **3.75**  **0.37** | **3.81**  **0.33** | **2.36**  **0.26** | **3.87**  **0.36** |
> |         |                    |                    |                    |                    |                    |                    |
>
> The results in the above Table show that the intricately structured CNN-RNN and Bayesian framework in cST-ML [1] led to an 8\% average reduction in QICE compared to the VAE model. DAC-ML [2], which builds upon cST-ML by incorporating model adaptation capabilities, achieved a 15\% reduction in QICE relative to the VAE-based approach. However, it's noteworthy that the proposed TMDM still surpasses both cST-ML [1] and DAC-ML [2] in performance, despite their enhancements over the standard VAE. The reasons for this significant outperformance by TMDM are detailed as follows:
>
> 1. Transformer-based architectures have shown remarkable success across a wide range of tasks. However, models like DAC-ML and cST-ML, which are based on CNN-RNN structures, face challenges in capturing long-term correlations. In the field of time series forecasting, specialized transformers are developed to address the nuances of multivariate time series. These specialized transformers often outperform generic transformer models. TMDM leverages these specifically tailored transformers, thereby gaining a significant edge over RNN-based methodologies.
>
> 2. The diffusion model, recognized as a state-of-the-art deep generative model, has been extensively validated for its enhanced generative capabilities over basic VAEs in numerous studies. TMDM integrates a conditional diffusion generative process, which facilitates accurate distribution forecasting in multivariate time series. This attribute equips TMDM with a substantial advantage in performance over other VAE-based models like cST-ML.
>
> 3. TMDM effectively utilizes transformers to distill crucial insights from historical time series data. These insights serve as prior knowledge, enabling the model to capture covariate-dependence effectively during both the forward and reverse phases of the diffusion process. This synergistic integration of transformers and diffusion models positions TMDM at the forefront in probabilistic multivariate time series forecasting.
>
> 4. TMDM's design transcends a mere combination of two modules. Instead, it integrates these models within a cohesive Bayesian framework, employing a hybrid optimization strategy. As detailed in Equation 15 of our paper, the first term guides the denoising model to predict uncertainty while subtly adjusting the conditional generative model to provide a more appropriate conditional representation. The second term facilitates the generation of improved conditional representations by harnessing the capabilities of a well-designed transformer. The two parts of the model adapt to each other within a hybrid framework, further enhancing the model's performance.
>
> [1] Zhang, Y., Li, Y., Zhou, X., \& Luo, J. (2020, November). cST-ML: Continuous spatial-temporal meta-learning for traffic dynamics prediction. In 2020 IEEE International Conference on Data Mining (ICDM), pp. 1418-1423.
>
> [2] Zhang, X., Li, Y., Zhou, X., Mangoubi, O., Zhang, Z., Filardi, V., \& Luo, J. (2021, December). DAC-ML: domain adaptable continuous meta-learning for urban dynamics prediction. In 2021 IEEE International Conference on Data Mining (ICDM), pp. 906-915.

---

> ### Author Response · Authors · 2023-11-18
>
> **Q2:**
>
> Thank you for your insightful question. Evaluating model performance with time-series data at different granularities is of significant importance in real-world applications. We believe TMDM can excel in such scenarios for the following reasons:
>
> 1. Similar to most time series forecasting models [4,5,6], we incorporate the actual timestamps as learnable time embeddings for each data point. Leveraging a well-designed Transformer, we can effectively capture the temporal correlations within the data. This design ensures that TMDM can adapt to various granularities of time-series data.
>
> 2. Within the diffusion model component, we also account for the time embedding in the data. This allows the model to generate multivariate time series with information from these embeddings, accommodating time series data at different granularities.
>
> 3. As shown in Table 1 in our paper, the selected datasets covered different granularities ranging from 10 minutes to 1 day, and TMDM demonstrated competitive performance across all databases. This confirms the model's ability to handle situations where time-series data is available at different granularities.
>
> 4. To further evaluate TMDM's ability to handle varying time-series granularities, we conducted an experiment where we randomly removed $D$ data points from a given time series $y_{0:M}$ and used this modified dataset to test TMDM with the same settings as described in the paper. This challenging experiment simulates scenarios where the time intervals in multivariate time series vary, making it a rigorous test of the model's performance under changing granularities.
>
> |         |                |            |             |            |            |
> |:-------:|:--------------:|:----------:|:-----------:|:----------:|:----------:|
> | Dataset |    Exchange    |   ETTm2    | Electricity |  Traffic   |   Wether   |
> | Metric  |   QICE  CRPS   | QICE  CRPS | QICE  CRPS  | QICE  CRPS | QICE  CRPS |
> | TMDM-60 |   4.78  0.32   | 3.72  0.39 | 3.73  0.31  | 2.36  0.28 | 3.91  0.37 |
> | TMDM-30 |   4.14  0.30   | 3.93  0.38 | 3.85  0.35  | 2.34  0.25 | 3.79  0.34 |
> |  TMDM   |   4.38  0.32   | 3.75  0.37 | 3.81  0.33  | 2.36  0.26 | 3.87  0.36 |
> |         |                |            |             |            |            |
>
> In the presented table, TMDM-60 refers to our model with a prediction length of 192 + 60, where we randomly exclude 60 samples from the data, introducing random time intervals between consecutive points. In this scenario, TMDM is tasked with forecasting multivariate time series (MTS) with varying granularities based on temporal embeddings. Remarkably, the results from the 30 and 60 settings display similar scores to the original setting, showcasing the efficacy of TMDM in handling time-series data with diverse granularities.
>
> [4] Liu, Y., Wu, H., Wang, J., \& Long, M. (2022). Non-stationary transformers: Exploring the stationarity in time series forecasting. Advances in Neural Information Processing Systems, 35, pp. 9881-9893.
>
> [5] Wu, H., Xu, J., Wang, J., \& Long, M. (2021). Autoformer: Decomposition transformers with auto-correlation for long-term series forecasting. Advances in Neural Information Processing Systems, 34, pp. 22419-22430.
>
> [6] Zhou, H., Zhang, S., Peng, J., Zhang, S., Li, J., Xiong, H., \& Zhang, W. (2021, May). Informer: Beyond efficient transformer for long sequence time-series forecasting. In Proceedings of the AAAI conference on artificial intelligence, Vol. 35, No. 12, pp. 11106-11115.

---

> > ### Author Response · Authors · 2023-11-22
> >
> > Dear Reviewer YJEP,
> >
> > We appreciate it if you could let us know whether our responses are able to address your concerns. We're happy to address any further concerns. Thank you,
> >
> > Best wishes.
> >
> > Paper2407 Authors

---

> > ### Comment · Reviewer_YJEP · 2023-11-22
> >
> > Thanks for the reply. My concerns have been addressed.

---

> > > ### Author Response · Authors · 2023-11-23
> > >
> > > Dear Reviewer YJEP,
> > >
> > > We are pleased to have addressed your concerns. In light of the changes made, could you kindly reconsider your score?
> > >
> > > Thank you for your time and consideration.
> > >
> > > Paper2407 Authors

---

### Meta-Review · Area_Chair_gwdi · 2023-12-06

**Metareview:**

This paper proposes a probabilistic forecasting method which incorporates transformers into the conditional diffusion process of diffusion models for forecasting. Although incorporating transformers into diffusion models is not particularly novel, the paper shows that the proposed method can deliver consistently good results even when compared against strong baselines. This work has potential to advance the state of the art and hence is worth publishing, but further revising the paper to address the outstanding concerns of some reviewers will make it appeal better to the community.

**Justification For Why Not Higher Score:**

It is not good enough to be accepted as a spotlight paper.

**Justification For Why Not Lower Score:**

The merits as highlighted above make it acceptable for publication, but it is fine with me not to accept it if there are many stronger papers.

---

### Decision · Program_Chairs · 2024-01-16

Accept (poster)